# Reasoning by Superposition: A Theoretical Perspective on Chain of Continuous Thought

**Hanlin Zhu***
UC Berkeley
hanlinzhu@berkeley.edu

**Shibo Hao***
UCSD
s5hao@ucsd.edu

**Zhiting Hu**
UCSD
zhh019@ucsd.edu

**Jiantao Jiao**
UC Berkeley
jiantao@berkeley.edu

**Stuart Russell**
UC Berkeley
russell@cs.berkeley.edu

**Yuandong Tian**
Meta AI
yuandong@meta.com

## Abstract

Large Language Models (LLMs) have demonstrated remarkable performance in many applications, including challenging reasoning problems via chain-of-thought (CoT) techniques that generate "thinking tokens" before answering the questions. While existing theoretical works demonstrate that CoT with discrete tokens boosts the capability of LLMs, recent work on continuous CoT lacks a theoretical understanding of why it outperforms discrete counterparts in various reasoning tasks, such as directed graph reachability, a fundamental graph reasoning problem that includes many practical domain applications as special cases. In this paper, we prove that a two-layer transformer with $D$ steps of continuous CoT can solve the directed graph reachability problem, where $D$ is the diameter of the graph, while the best known result of constant-depth transformers with discrete CoT requires $O(n^2)$ decoding steps where $n$ is the number of vertices ($D < n$). In our construction, each continuous thought vector is a superposition state that encodes multiple search frontiers simultaneously (i.e., *parallel breadth-first search (BFS)*), while discrete CoT must choose a single path sampled from the superposition state, which leads to a sequential search that requires many more steps and may be trapped in local solutions. We also performed extensive experiments to verify that our theoretical construction aligns well with the empirical solution obtained via training dynamics. Notably, encoding of multiple search frontiers as a superposition state automatically *emerges* in training continuous CoT, without explicit supervision to guide the model to explore multiple paths simultaneously. Our code is available at https://github.com/Ber666/reasoning-by-superposition.

## 1 Introductions

Large language models (LLMs) have shown strong performance in many reasoning tasks, especially when empowered with chain-of-thought (CoT) [Wei et al., 2022] (e.g., hard problems like AIME and math proving). However, they also struggle with tasks that require more sophisticated reasoning capability [Kambhampati, 2024], e.g., reasoning and planning problems of increasing scales [Zheng et al., 2024, Xie et al., 2024], even with CoT [Valmeekam et al., 2024, Zhou et al., 2025].

It remains an open problem how to expand existing discrete CoT to solve more complex reasoning problems. Recently, Hao et al. [2024] proposes COCONUT (chain-of-continuous-thought) that uses

---

*Equal contributions.

39th Conference on Neural Information Processing Systems (NeurIPS 2025).

continuous latent thoughts for reasoning, showing empirical performance boost on synthetic tasks such as directed graph reachability (i.e., given a specification of a directed graph and one starting node, determine which candidate destination node is reachable), as well as strong performance on real-world math reasoning benchmarks such as GSM8K [Cobbe et al., 2021]. Interestingly, COCONUT shows preliminary results that continuous latent thought may store multiple candidate search frontiers simultaneously, before the final answer is reached. This is in sharp contrast with discrete CoT, in which each discrete thought token has to be sampled (or "realized") before feeding into LLMs in an autoregressive manner. However, the expressive power and the mechanism of continuous thought still remain elusive and lack a deep understanding.

In this work, we explore the mechanism of COCONUT for the problem of *graph reachability*, i.e., whether there exists a path from given start and end nodes in a directed graph. The problem setting is general [Ye et al., 2024, Hao et al., 2024, Zhou et al., 2025] and includes many important theoretical problems (e.g., Turing machine halting problem) and practical use cases (e.g., knowledge graph). Given this setting, we proved that a two-layer transformer with $D$ steps of continuous thought can solve graph reachability for graphs of $n$ vertices, where $D < n$ is the graph's diameter (longest path length between two nodes). In contrast, for graph reachability, the best existing result on constant-depth transformers with discrete CoT requires $O(n^2)$ steps [Merrill and Sabharwal, 2023a].

Intuitively, in our construction, each latent thought vector is a superposition of multiple valid search traces, and thus can perform *implicit parallel search* on the graph in each autoregressive step. The continuous thoughts can be regarded as "superposition states" in quantum mechanics [Böhm, 2013], storing multiple search frontiers simultaneously, and thus enabling efficient breadth-first search (BFS). In contrast, discrete thought tokens can be viewed as "collapsed states" from superpositions. This forces the model to choose a branch deterministically, yielding either an incorrect greedy search or a depth-first style search with backtracking, which requires more computation. Unlike previous theoretical work that constructs positional encodings specifically for a given problem or even for a given input length, our construction works for widely-used positional encodings in practice, such as sinusoidal positional encoding [Vaswani et al., 2017] and rotary position embedding [Su et al., 2024].

Moreover, we show that our theoretical construction can be achieved in gradient-based training. Specifically, a two-layer transformer with continuous CoT outperforms a 12-layer one with discrete CoT on graph reachability. An inspection of attention patterns and their underlying representation demonstrates that the continuous thought indeed encodes multiple plausible search frontiers in parallel in superposition states. Notably, such a superpositional representation automatically *emerges* from training only with the optimal path of graph reachability, without strong supervision that aligns the latent thought vectors with other plausible search traces.

## 1.1 Related works

**LLM reasoning in text and latent spaces.** LLM's reasoning capability can be significantly boosted by chain-of-thought (CoT) [Wei et al., 2022], which allows LLMs to explicitly output intermediate thoughts in text space before predicting the final answer. CoT includes prompt-only methods [Khot et al., 2022, Zhou et al., 2022] and training with samples containing intermediate thoughts [Yue et al., 2023, Yu et al., 2023, Wang et al., 2023b, Shao et al., 2024]. Besides text-based CoT, many previous works also study LLM reasoning in the latent space [Goyal et al., 2023, Wang et al., 2023c, Pfau et al., 2024, Su et al., 2025] where the intermediate thoughts do not necessarily correspond to textual tokens. In particular, Hao et al. [2024] proposed to train LLMs to reason in a continuous latent space, which outperforms discrete CoT on graph reasoning tasks, especially for graphs with high branching factors. Based on empirical case studies in Hao et al. [2024], continuous thoughts are hypothesized to encode multiple plausible search frontiers simultaneously. In this work, we formally study the mechanism and theoretically show that transformers equipped with continuous thoughts benefit from superposition states during reasoning.

**Expressivity of transformers.** There is a long line of work studying the expressivity of transformers [Yun et al., 2019, Bhattamishra et al., 2020a,b, Pérez et al., 2021, Likhosherstov et al., 2021, Yao et al., 2021, Edelman et al., 2022, Akyürek et al., 2022, Merrill and Sabharwal, 2023b]. A more recent line of work shows CoT can improve the expressivity of transformers [Liu et al., 2022, Feng et al., 2023, Merrill and Sabharwal, 2023a, Li et al., 2024]. For example, Liu et al. [2022] studies low-depth transformer expressivity for semi-automata, of which the setting corresponds to

one CoT step. Feng et al. [2023] shows that constant-depth transformers with CoT can solve certain P-complete problems. Li et al. [2024] further provides constructions of constant-depth transformer for each problem in P/poly with CoT. Merrill and Sabharwal [2023a] studies the expressivity with different lengths of CoT, showing that logarithmic steps of CoT in input length can expand the upper bound of constant-depth transformer expressivity from $\mathsf{TC}^0$ to $\mathsf{L}$, while a linear number of steps can further expand the upper bound to $\mathsf{NC}^1$-complete. While these expressivity results mainly focus on discrete CoT, theoretical studies on continuous CoT [Hao et al., 2024] are rare, which our work is focused on. Gozeten et al. [2025] studies the expressivity of one-layer transformers with continuous CoT on the minimum non-negative sum problem, demonstrating the superposition mechanism in the arithmetic domain, which complements our results on the graph reachability problem. While their construction requires an exponentially large embedding dimension, our theoretical construction requires only a linear embedding dimension in the graph size. Moreover, unlike many previous works that construct problem-specific (or even length-specific) positional encodings, our construction applies to practical positional encodings, such as sinusoidal [Vaswani et al., 2017] and RoPE [Su et al., 2024].

**Reasoning as graph problems.**    Graph problems are essential to understand LLM reasoning capability since many reasoning problems can be abstracted as computational graphs [Ye et al., 2024, Zhou et al., 2025] where relational data fed into transformers can be modeled as edges [Wang et al., 2024a,b, Guo et al., 2025]. Many previous works have shown that pretrained LLMs can deal with reasoning tasks in graphs, but may still have difficulties with more complicated tasks [Wang et al., 2023a, Guo et al., 2023, Fatemi et al., 2023, Sanford et al., 2024, Luo et al., 2024, Dai et al., 2024, Cohen et al., 2025]. Other works study how transformers solve classic and fundamental graph problems, such as graph reachability [Merrill and Sabharwal, 2023a, 2025], shortest path [Cohen et al., 2025], etc. For example, Cohen et al. [2025] shows that a two-layer transformer can leverage spectral decomposition of the line graph to predict the shortest path on small-scale undirected graphs. Merrill and Sabharwal [2025] shows that a log-depth transformer can solve directed graph reachability, which constant-depth transformers can not solve. For a constant-depth transformer, Merrill and Sabharwal [2023a] shows directed graph reachability can be solved with $O(n^2)$ CoT steps where $n$ is the number of vertices, while it remains unclear whether a smaller number of discrete CoT steps can solve the task. On the contrary, our work shows that a two-layer transformer can solve graph reachability for a $D$-diameter graph with $D$ continuous thought steps.

## 2   Preliminaries

**Basic notations.**    For any integer $N > 0$, we use $[N]$ to denote the set $\{1, 2, \ldots, N\}$. For any finite set $\mathcal{X}$, let $|\mathcal{X}|$ denote the cardinality of $\mathcal{X}$. Let $\mathbb{R}$ be the set of real numbers. We use $\mathbb{1}\{\cdot\}$ to denote the indicator function. Without further clarification, we use lower-case bold letters (e.g., $\boldsymbol{x}, \boldsymbol{\theta}$) and upper-case bold letters (e.g., $\mathbf{W}, \mathbf{U}$) to denote vectors and matrices, respectively. In particular, we use $\mathbf{I}_d$ to denote a $d \times d$ identity matrix, use $\mathbf{0}_{m \times n}$ (or $\mathbf{0}_m$) to denote an $m \times n$ zero matrix (or an $m$-dimensional zero vector), and use $\boldsymbol{e}_i$ to denote a one-hot vector of which the $i$-th entry is one and other entries are all zero, where the dimension of $\boldsymbol{e}_i$ can be inferred from the context. We also use $\|\cdot\|_\infty$ and $\|\cdot\|_2$ to represent $L_\infty$ norm and $L_2$ norm of vectors, respectively. For vectors $\mathbf{u} \in \mathbb{R}^m$ and $\mathbf{v} \in \mathbb{R}^n$, let $\mathbf{u} \otimes \mathbf{v} = \mathbf{u}\mathbf{v}^\top \in \mathbb{R}^{m \times n}$ denote their outer product. Also, for vectors $\mathbf{u}, \mathbf{v} \in \mathbb{R}^d$, let $\langle \mathbf{u}, \mathbf{v} \rangle = \mathbf{u}^\top \mathbf{v}$ denote their inner product. For any vector $\boldsymbol{x} = (x_1, \ldots, x_d) \in \mathbb{R}^d$, we define the softmax function $\mathsf{SoftMax} : \mathbb{R}^d \to \mathbb{R}^d$ as $\mathsf{SoftMax}(\boldsymbol{x})_i = \exp(x_i)/(\sum_{j=1}^d \exp(x_j))$, and the layer normalization operator $\mathsf{LayerNorm}(\boldsymbol{x}) = \boldsymbol{x}/\|\boldsymbol{x}\|_2$. Moreover, for a sequence of vectors $(\boldsymbol{x}_1, \boldsymbol{x}_2, \ldots, \boldsymbol{x}_t) \in \mathbb{R}^{d \times t}$, we abuse the notation $\mathsf{LayerNorm}(\boldsymbol{x}_1, \ldots, \boldsymbol{x}_t) = (\mathsf{LayerNorm}(\boldsymbol{x}_1), \ldots, \mathsf{LayerNorm}(\boldsymbol{x}_t)) \in \mathbb{R}^{d \times t}$.

**Tokens and embeddings.**    For a fixed positive integer $V > 0$, let $\mathsf{Voc} = [V]$ denote a vocabulary of size $V$. For each token $v \in \mathsf{Voc}$, there is an associated token embedding $\mathbf{u}_v \in \mathbb{R}^d$ where $d > 0$ is the embedding dimension. Assume $d = 3d_{\mathsf{TE}} + d_{\mathsf{PE}}$. We refer to the first $d_{\mathsf{TE}}$ entries of a $d$-dimensional vector as its content, the subsequent $d_{\mathsf{TE}}$ entries as its first buffer, the following $d_{\mathsf{TE}}$ entries as its second buffer, and the final $d_{\mathsf{PE}}$ entries as its effective positional encoding. Formally, for a vector $\boldsymbol{x} = (x_1, x_2, \ldots, x_d)^\top \in \mathbb{R}^d$, we define $\mathsf{content}(\boldsymbol{x}) = (x_1, \ldots, x_{d_{\mathsf{TE}}})^\top$, $\mathsf{buffer}_1(\boldsymbol{x}) = (x_{d_{\mathsf{TE}}+1}, \ldots, x_{2d_{\mathsf{TE}}})^\top$, $\mathsf{buffer}_2(\boldsymbol{x}) = (x_{2d_{\mathsf{TE}}+1}, \ldots, x_{3d_{\mathsf{TE}}})^\top$ and $\mathsf{pos}(\boldsymbol{x}) = (x_{3d_{\mathsf{TE}}+1}, \ldots, x_d)^\top$. Let $\tilde{\mathbf{u}}_v = \mathsf{content}(\mathbf{u}_v) \in \mathbb{R}^{d_{\mathsf{TE}}}$ for any $v \in \mathsf{Voc}$. Assume for each $\mathbf{u}_v$, only the first $d_{\mathsf{TE}}$ entries are

non-zero. Furthermore, let $\mathbf{U} = [\tilde{\mathbf{u}}_1, \tilde{\mathbf{u}}_2, \ldots, \tilde{\mathbf{u}}_V] \in \mathbb{R}^{d_{\mathsf{TE}} \times V}$ and we assume that $\mathbf{U}^\top \mathbf{U} = \mathbf{I}_V$, i.e., the token embeddings are orthonormal.

---

**Algorithm 1** Transformer (TF)

---

**Input:** Parameters $\theta = (\theta_{\mathsf{PE}}, \{\theta_{\mathsf{Attn}}^{(l,h)}\}_{l=0,h=0}^{L-1,H_l-1}, \{\theta_{\mathsf{MLP}}^{(l)}\}_{l=0}^{L-1})$, input embeddings $\mathbf{h} = (\mathbf{h}_1, \ldots, \mathbf{h}_t)$

1: $\mathbf{h}_i^{(0)} \leftarrow \mathbf{h}_i + \mathrm{PosEncode}_{\theta_{\mathsf{PE}}}(i), \quad \forall i \in [t]$          ▷ adding positional encoding
2: **for** $l = 0$ to $L - 1$ **do**
3:      $\mathbf{h}^{(l+0.5)} \leftarrow \mathbf{h}^{(l)} + \sum_{h=0}^{H_l-1} \mathsf{Attn}_{\theta_{\mathsf{Attn}}^{(l,h)}}(\mathbf{h}^{(l)})$          ▷ self attention
4:      $\mathbf{h}^{(l+1)} \leftarrow \mathsf{LayerNorm}\left(\mathsf{MLP}_{\theta_{\mathsf{MLP}}^{(l)}}(\mathbf{h}^{(l+0.5)})\right)$          ▷ MLP and layer normalization
5: **end for**
**Output:** Embedding of the last layer at the last position $\mathbf{h}_t^{(L)}$.

---

**Transformer architectures.** An $L$-layer autoregressive transformer receives a sequence of input embeddings $\mathbf{h} = \mathbf{h}_{[t]} \triangleq (\mathbf{h}_1, \mathbf{h}_2, \ldots, \mathbf{h}_t) \in \mathbb{R}^{d \times t}$ and outputs $\mathsf{TF}_\theta(\mathbf{h}) \in \mathbb{R}^d$ where $\mathsf{TF}_\theta(\cdot)$ is defined in Algorithm 1. Let $\mathbf{W}_{\mathbf{O}} \in \mathbb{R}^{V \times d}$ be the decoding matrix. A traditional decoder will sample the next token $v_{t+1} \sim \mathsf{SoftMax}(\mathbf{W}_{\mathbf{O}}\mathsf{TF}_\theta(\mathbf{h}))$ and append its token embedding in position $t + 1$, i.e., $\mathbf{h}_{t+1} = \mathbf{u}_{v_{t+1}}$, to autoregressively generate subsequent outputs. When using the chain of continuous thought [Hao et al., 2024], one skips the sampling step and directly appends the output of the transformer as the input embedding of the next position, i.e., $\mathbf{h}_{t+1} = \mathsf{TF}_\theta(\mathbf{h})$. The parameter $\theta$ contains positional encodings $\theta_{\mathsf{PE}}$, $L$ attention layers where each layer $l$ contains $H_l$ heads $\{\theta_{\mathsf{Attn}}^{(l,h)}\}_{l=0,h=0}^{L-1,H_l-1}$ and $L$ MLP layers $\{\theta_{\mathsf{MLP}}^{(l)}\}_{l=0}^{L-1}$. The definitions of attention heads and MLPs are in Algorithm 2.

---

**Algorithm 2** Causal Self-Attention (Attn) and (position-wise) Multilayer Perceptron (MLP)

---

**Input:** $\theta_{\mathsf{Attn}} = (\mathbf{Q}, \mathbf{K}, \mathbf{V}, \mathbf{O}), \theta_{\mathsf{MLP}} = (\{\mathbf{W}_i\}_{i=1}^{L_{\mathsf{MLP}}}, \{\sigma_i(\cdot)\}_{i=1}^{L_{\mathsf{MLP}}-1})$, input $\mathbf{h} = (\mathbf{h}_1, \ldots, \mathbf{h}_t)$

1: $\mathbf{q}_i \leftarrow \mathbf{Q}\mathbf{h}_i, \quad \mathbf{k}_i \leftarrow \mathbf{K}\mathbf{h}_i, \quad \mathbf{v}_i \leftarrow \mathbf{V}\mathbf{h}_i, \quad \forall i \in [t]$      ▷ compute queries, keys, values
2: **for** $i = 1$ to $t$ **do**
3:      $s_i \leftarrow \mathsf{SoftMax}(\langle \mathbf{q}_i, \mathbf{k}_1 \rangle, \ldots, \langle \mathbf{q}_i, \mathbf{k}_i \rangle), \quad \mathbf{h}_i^{\mathsf{Attn}} \leftarrow \mathbf{O} \sum_{j=1}^{i} s_{i,j} \mathbf{v}_j$
4:      $\mathbf{h}_i^{\mathsf{MLP}} \leftarrow \mathbf{W}_{L_{\mathsf{MLP}}} \sigma_{L_{\mathsf{MLP}}-1}(\cdots \mathbf{W}_2 \sigma_1(\mathbf{W}_1 \mathbf{h}_i) \cdots)$
5: **end for**
**Output:** $\mathsf{Attn}_{\theta_{\mathsf{Attn}}}(\mathbf{h}) = (\mathbf{h}_1^{\mathsf{Attn}}, \ldots, \mathbf{h}_t^{\mathsf{Attn}})$ or $\mathsf{MLP}_{\theta_{\mathsf{MLP}}}(\mathbf{h}) = (\mathbf{h}_1^{\mathsf{MLP}}, \ldots, \mathbf{h}_t^{\mathsf{MLP}})$

---

**Positional encodings.** Given an input sequence $(\mathbf{h}_1, \ldots, \mathbf{h}_T)$, for each position $i \in [T]$, there is a corresponding positional encoding $\mathbf{p}_i = \mathrm{PosEncode}_{\theta_{\mathsf{PE}}}(i) \in \mathbb{R}^d$. For each $\mathbf{p}_i$, we assume that only the last $d_{\mathsf{PE}}$ entries are non-zero and thus call $d_{\mathsf{PE}}$ the effective positional encoding dimension. For notation convenience, we denote $\tilde{\mathbf{h}}_i = \mathrm{content}(\mathbf{h}_i) \in \mathbb{R}^{d_{\mathsf{TE}}}, \bar{\mathbf{p}}_i = \mathrm{pos}(\mathbf{p}_i) \in \mathbb{R}^{d_{\mathsf{PE}}}$ for any $i \in [T]$. We use the widely used sinusoidal positional encoding for $\bar{\mathbf{p}}_i = (\bar{p}_{i,1}, \ldots, \bar{p}_{i,d_{\mathsf{PE}}})$ as defined below.

**Definition 1** (Positional Encoding). *Let $d_{\mathsf{PE}}$ be even. We use positional encoding generated by sinusoidal functions Vaswani et al. [2017]. Specifically, for any position $i \geq 1$ and any index $j \in [d_{\mathsf{PE}}/2]$, we have*

$$\bar{p}_{i,2j-1} = \cos(i \cdot M^{-2j/d_{\mathsf{PE}}}), \quad \bar{p}_{i,2j} = \sin(i \cdot M^{-2j/d_{\mathsf{PE}}}),$$

*where $M > 0$ is a large constant integer, e.g., $M = 10^4$ as chosen in Vaswani et al. [2017].*

**Remark 1.** *We also discuss theoretical construction with RoPE[Su et al., 2024] (Appendix B.6).*

## 3 Problem Formulations

**Graph reachability.** Let $\mathcal{G} = (\mathcal{V}, \mathcal{E})$ be a directed graph, where $\mathcal{V} = \{v_1, v_2, \ldots, v_n\}$ is the set of vertices and $\mathcal{E} = \{e_1, e_2, \ldots, e_m\}$ is the set of edges. Each vertex $v_i \in \mathcal{V}$ corresponds to a token in the vocabulary, and thus we use $v_i$ to represent both a vertex and its corresponding token. Let

$n_{\max} > 0$ denote the maximum possible number of vertices of a graph. Note that $n_{\max} < |\mathsf{Voc}|$. Each edge $e_i \in \mathcal{E}$ is a tuple, where $e_i = (\mathsf{s}_i, \mathsf{t}_i) \in \mathcal{V} \times \mathcal{V}$ denotes there is a directed edge from the source node $\mathsf{s}_i$ to the target node $\mathsf{t}_i$. Given a graph (i.e., all the edges of the graph), two candidate destination nodes $\mathsf{c}_1$ and $\mathsf{c}_2$, and a root node $\mathsf{r}$, the task is to identify which of the two nodes can be reached by $\mathsf{r}$. Note that we guarantee one and only one of $\mathsf{c}_1$ and $\mathsf{c}_2$ is reachable by $\mathsf{r}$, which we denote by $\mathsf{c}_{i^*}$.

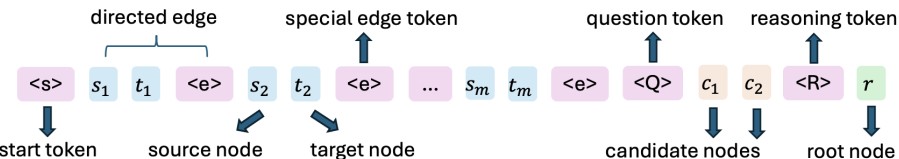

Figure 1: Prompt format of the graph reachability problem.

**Input structures.** The prompt structure is illustrated in Figure 1. The prompt starts with the BOS (beginning of sentence) token ``. The subsequent $3m$ tokens represent $m$ edges, where each edge is represented by the source node $\mathsf{s}_i$, target node $\mathsf{t}_i$, and a special edge token `<e>` that marks the end of an edge. Then there is a special question token `<Q>` followed by two candidate destination nodes $\mathsf{c}_1$ and $\mathsf{c}_2$. Finally, there is a special reasoning token `<R>` followed by a root node $\mathsf{r}$. See Table 2 for the full list of token notations. Let $t_0 = 3m + 6$ be the length of the prompt, and let $(\mathbf{h}_1, \mathbf{h}_2, \ldots \mathbf{h}_{t_0})$ be the input embedding sequence where $\mathbf{h}_i$ is the token embedding of the $i$-th token in the prompt.

**Chain of continuous thought.** We allow transformers to utilize continuous thoughts. Concretely, for $c = 1, 2, \ldots$, the transformer autoregressively generates $\mathbf{h}_{t_0+c} = \mathsf{TF}_\theta(\mathbf{h}_1, \ldots, \mathbf{h}_{t_0+c-1})$. For notation convenience, we use $[\mathsf{t}_c] = \mathbf{h}_{t_0+c}$ to represent the continuous thought of step $c$ for $c \geq 0$, and thus $[\mathsf{t}_0] = \mathbf{u}_\mathsf{r}$. To request the transformer to make the final prediction after $C$ steps of thoughts, one simply appends a special answer token `<A>` at the end of the sequence, i.e., sets $\mathbf{h}_T = \mathbf{u}_{\texttt{<A>}}$ where $T = t_0 + C + 1$, and gets the prediction sampled from $\mathsf{SoftMax}(\mathbf{W_O}\mathsf{TF}_\theta(\mathbf{h}_{[T]}))$ or using greedy decoding $\arg\max_{v \in \mathsf{Voc}} \mathbf{W_O}\mathsf{TF}_\theta(\mathbf{h}_{[T]})$. We denote $\widetilde{\mathsf{TF}}_{\theta,C,\mathbf{W_O}}(\mathbf{h}_{[t_0]}) = \arg\max_{v \in \mathsf{Voc}} \mathbf{W_O}\mathsf{TF}_\theta(\mathbf{h}_{[T]})$ as the output token of greedy decoding after generating $C$ steps of continuous thoughts.

**Position index.** To present the position of each token or continuous thought in the sequence in a clear way, we use $\mathsf{Idx}(v)$ to represent the position of a token in the input sequence (e.g., $\mathsf{Idx}(\texttt{}) = 1, \mathsf{Idx}(\mathsf{s}_i) = 3i - 1, \mathsf{Idx}(\texttt{<Q>}) = 3m + 2$), use $\mathsf{Idx}(\texttt{<e>}, i) = 3i + 1$ to represent the position of the $i$-th `<e>` token in the prompt, and use $\mathsf{Idx}([\mathsf{t}_i]) = t_0 + i$ to represent the position of the continuous thought of step $i$. See Table 3 for the complete list of position indices.

In the following sections, we demonstrate that the chain of continuous thought can efficiently solve the graph reachability problem both theoretically (Section 4) and empirically (Section 5).

# 4 Theoretical Results

In this section, we theoretically prove that a two-layer transformer with continuous thought can efficiently solve the graph reachability problem. We first introduce a basic building block, the *attention chooser*, in our transformer constructions in Section 4.1. Then we present the key result that continuous thought maintains a superposition state of multiple search traces simultaneously in Section 4.2. We show our main theorem in Section 4.3 and make further discussion in Section 4.4.

## 4.1 Attention chooser

We use the attention chooser as a building block in our construction, which will choose the appropriate positions to attend conditioned on the token in the current position. This allows us to use the same parameter constructions for various input lengths. The proof is deferred to Appendix B.1.

**Lemma 1** (Attention chooser, informal version of Lemma 3). *Under sinusoidal positional encoding as defined in Definition 1, for any token* `<x>` $\in \mathsf{Voc}$ *and relative position* $\ell \geq 0$*, there exists*

*a construction of* $\mathbf{K}, \mathbf{Q} \in \mathbb{R}^{(2d_{\mathsf{PE}}) \times d}$, *such that for any input sequence* $\mathbf{h}_{[T]}$ *that satisfying mild assumptions (see Lemma 3 in Appendix B.2), it holds that for any position* $i \in [T]$, *it will pay almost all attention to position* $(i - \ell)$ *if* $\mathbf{h}_i = \mathbf{u}_{\texttt{<x>}}$, *and pay most attention to position one otherwise.*

*Proof sketch.* We define vector $\tilde{\mathbf{u}}_{\overline{\texttt{<x>}}} = \sum_{v \in \mathsf{Voc} \setminus \{\texttt{<x>}\}} \tilde{\mathbf{u}}_v \in \mathbb{R}^{d_{\mathsf{TE}}}$ as the superposition of all token embeddings in the vocabulary except for $\texttt{<x>}$. By the property of sinusoidal positional encoding, there exists a rotation matrix $\mathbf{R}^{(\ell)}$ as in Lemma 4 in Appendix B.5, s.t. $\bar{\mathbf{p}}_{i+\ell} = \mathbf{R}^{(\ell)}\bar{\mathbf{p}}_i, \ \forall i \geq 1$. Then we can construct the query and key matrices as

$$\mathbf{Q} = \begin{bmatrix} \mathbf{0}_{d_{\mathsf{PE}} \times d_{\mathsf{TE}}} & \mathbf{0}_{d_{\mathsf{PE}} \times 2d_{\mathsf{TE}}} & \mathbf{I}_{d_{\mathsf{PE}}} \\ \xi \bar{\mathbf{p}}_1 \otimes \tilde{\mathbf{u}}_{\overline{\texttt{<x>}}} & \mathbf{0}_{d_{\mathsf{PE}} \times 2d_{\mathsf{TE}}} & \mathbf{0}_{d_{\mathsf{PE}} \times d_{\mathsf{PE}}} \end{bmatrix}, \quad \mathbf{K} = \begin{bmatrix} \mathbf{0}_{d_{\mathsf{PE}} \times 3d_{\mathsf{TE}}} & \eta \mathbf{R}^{(\ell)} \\ \mathbf{0}_{d_{\mathsf{PE}} \times 3d_{\mathsf{TE}}} & \eta \mathbf{I}_{d_{\mathsf{PE}}} \end{bmatrix},$$

where $\xi, \eta > 0$ and thus the query and key vectors can be calculated as

$$\mathbf{q}_i = \mathbf{Q}(\mathbf{h}_i + \mathbf{p}_i) = \begin{bmatrix} \bar{\mathbf{p}}_i \\ \xi \langle \tilde{\mathbf{u}}_{\overline{\texttt{<x>}}}, \tilde{\mathbf{h}}_i \rangle \bar{\mathbf{p}}_1 \end{bmatrix}, \quad \mathbf{k}_i = \mathbf{K}(\mathbf{h}_i + \mathbf{p}_i) = \begin{bmatrix} \eta \mathbf{R}^{(\ell)} \bar{\mathbf{p}}_i \\ \eta \bar{\mathbf{p}}_i \end{bmatrix} = \begin{bmatrix} \eta \bar{\mathbf{p}}_{i+\ell} \\ \eta \bar{\mathbf{p}}_i \end{bmatrix}.$$

Now for any $1 \leq j \leq i \leq T$, we have $\langle \mathbf{q}_i, \mathbf{k}_j \rangle = \eta \left( \langle \bar{\mathbf{p}}_i, \bar{\mathbf{p}}_{j+\ell} \rangle + \xi \langle \tilde{\mathbf{u}}_{\overline{\texttt{<x>}}}, \tilde{\mathbf{h}}_i \rangle \langle \bar{\mathbf{p}}_1, \bar{\mathbf{p}}_j \rangle \right)$. Fix any $i \in [T]$. By the property of sinusoidal positional encoding, $\langle \bar{\mathbf{p}}_1, \bar{\mathbf{p}}_{i'} \rangle$ is maximized when $i' = i$ as in Lemma 5 in Appendix B.5. If $\mathbf{h}_i = \mathbf{u}_{\texttt{<x>}}$, it holds that $\langle \tilde{\mathbf{u}}_{\overline{\texttt{<x>}}}, \tilde{\mathbf{h}}_i \rangle = 0$ and thus $\langle \mathbf{q}_i, \mathbf{k}_j \rangle$ is determined by $\langle \bar{\mathbf{p}}_i, \bar{\mathbf{p}}_{j+\ell} \rangle$, which is maximized at $j = i - \ell$. If $\mathbf{h}_i \neq \mathbf{u}_{\texttt{<x>}}$, one can show that $\langle \tilde{\mathbf{u}}_{\overline{\texttt{<x>}}}, \tilde{\mathbf{h}}_i \rangle \geq 1$ and thus $\langle \mathbf{q}_i, \mathbf{k}_j \rangle$ is largely determined by $\langle \bar{\mathbf{p}}_1, \bar{\mathbf{p}}_j \rangle$ when $\xi$ is large, and thus maximized at $j = 1$. $\quad\square$

## 4.2 Continuous thought maintains superposition states

Recall that $\mathbf{h} = \mathbf{h}_{[t_0]}$ denotes the input sequence as defined in Section 3. We define $\mathcal{V}_c$ as the set of vertices that are reachable from $\mathbf{r}$ within $c$ steps. Below, we present our key lemma.

**Lemma 2** (Continuous thought maintains superposition states)**.** *We autoregressively generate* $[t_{c+1}] = \mathsf{TF}_\theta(\mathbf{h}_{[t_0]}, [t_1] \ldots, [t_c])$ *for any* $c \geq 0$. *There exists a construction of* $\theta$ *such that*

$$[t_c] = \mathbf{h}_{t_0+c} = \frac{1}{\sqrt{|\mathcal{V}_c|}} \sum_{v \in \mathcal{V}_c} \mathbf{u}_v, \tag{1}$$

*i.e., the* $c$-*th continuous thought is the normalized superposition of all vertices that can be reached from* $\mathbf{r}$ *within* $c$ *steps.*

Lemma 2 precisely characterizes that each continuous thought is a superposition of all reachable vertices. We provide a proof sketch below and defer the complete proof to Appendix B.2.

*Proof sketch.* We prove by induction. For $c = 0$, by definition, $\mathcal{V}_0 = \{\mathbf{r}\}$ and $[t_0] = \mathbf{u}_{\mathbf{r}} = \frac{1}{\sqrt{|\mathcal{V}_0|}} \sum_{v \in \mathcal{V}_0} \mathbf{u}_v$. Now we briefly show how to construct the two-layer transformer such that under the induction assumption that (1) holds for $0, \ldots, c$, we can obtain that (1) also holds for $c + 1$.

**First layer attention.** The first attention layer contains five attention heads, and each head is an attention chooser as constructed in Lemma 1. Let $h_k = (\texttt{<x>}, \ell)$ denote the $k$-th attention head that attends to position $(i - \ell)$ when the $i$-th token is $\texttt{<x>}$ and attends to the first token otherwise. We construct $h_0 = (\texttt{<e>}, 2), h_1 = (\texttt{<e>}, 1), h_2 = (\texttt{<R>}, 2), h_3 = (\texttt{<R>}, 1), h_4 = (\texttt{<A>}, 1)$, which is illustrated in Figure 2. For each head, the value matrix will read the value in the content space, and the output matrix will copy the value state to a designated space.

**Second layer attention.** In the second layer, we only need one attention head. Note that after the first layer, for the $i$-th edge token $\texttt{<e>}$, we have $\mathsf{buffer}_1(\mathbf{h}_{\mathsf{Idx}(\texttt{<e>},i)}) = \tilde{\mathbf{u}}_{\mathbf{s}_i}$ and $\mathsf{buffer}_2(\mathbf{h}_{\mathsf{Idx}(\texttt{<e>},i)}) = \tilde{\mathbf{u}}_{\mathbf{t}_i}$. By the induction assumption, the current thought $[t_c]$ is a superposition of all vertices in $\mathcal{V}_c$. We construct the query and key matrices such that $[t_c]$ pays large attention to the $i$-th edge token $\texttt{<e>}$ if $\mathbf{s}_i \in \mathcal{V}_c$ (roughly speaking, we can view $\mathbf{q}_{\mathsf{Idx}([t_c])} = [t_c], \mathbf{k}_{\mathsf{Idx}(\texttt{<e>},i)} = \mathbf{u}_{\mathbf{s}_i}$, and their inner product is positive iff $\mathbf{s}_i \in \mathcal{V}_c$), and add the target node $\mathbf{t}_i$ stored in buffer 2 back to the current thought (see Figure 3). This is exactly a one-step expansion of the currently explored vertices $\mathcal{V}_c$ and thus the continuous thought at the next step $(c + 1)$ will correspond to $\mathcal{V}_{c+1}$.

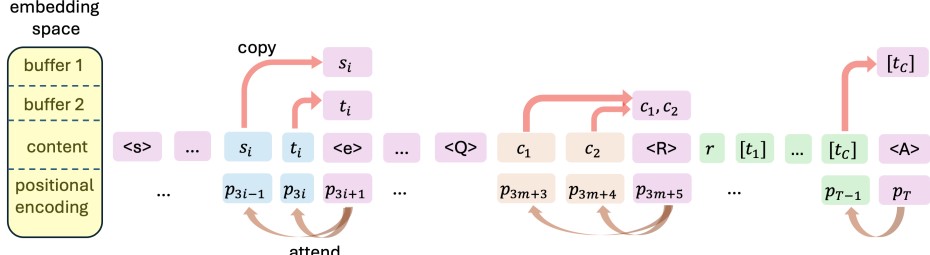

Figure 2: Illustration of the embedding space and first layer attention mechanism.

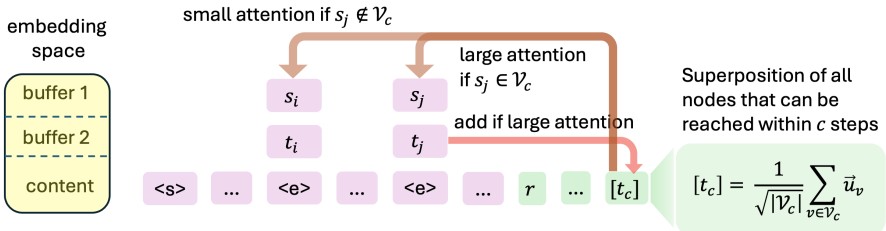

Figure 3: Illustration of the second layer attention mechanism for thought generation. We omit the positional encoding space since they are not used in the second layer.

**MLP as filter for signals.** Note that after the attention layer, the weight of each node in the current thought is not uniform, and the current thought might contain noise tokens since the normalized attention score to each position is non-zero. We use the MLP layer to filter out the noise token and adjust the weight of each node in $\mathcal{V}_{c+1}$. Informally, for a superposition state $\mathbf{h} = \sum_{v \in \mathsf{Voc}} \lambda_v \mathbf{u}_v$, we want to eliminate noise tokens $v$ where $\lambda_v < \varepsilon$, and want to equalize the weights of other tokens. By setting the first layer parameter as $\mathbf{W}_1 = [\mathbf{u}_1, \ldots, \mathbf{u}_V]^\top$, nonlinearity as $\sigma(x) = \mathbb{1}\{x \geq \varepsilon\}$ and second layer as $\mathbf{W}_2 = \mathbf{W}_1^\top$, we have $\mathbf{W}_2(\sigma(\mathbf{W}_1\mathbf{h})) = \sum_{v \in \mathsf{Voc}} \mathbb{1}\{\lambda_v \geq \varepsilon\}\mathbf{u}_v$, where $\mathbf{W}_1$ rotates the basis $\{\mathbf{u}_v\}$ to the standard basis $\{\mathbf{e}_v\}$, $\sigma(\cdot)$ serves as a coordinate-wise filter, and $\mathbf{W}_2$ rotates the basis back. After layer normalization, we can obtain that (1) also holds for $c+1$. □

### 4.3 Measuring the superposition state as prediction

Since the continuous thought $[\mathtt{t_c}]$ is a superposition of all vertices in $\mathcal{V}_c$, as long as $\mathtt{c}_{i*}$ can be reached by $\mathtt{r}$ within $C$ steps, the superposition state $[\mathtt{t_c}]$ will contain $\mathtt{c}_{i*}$. At the final prediction step, the answer token <A> will "measure" the superposition state $[\mathtt{t_c}]$ using $\mathtt{c}_1$ and $\mathtt{c}_2$ and predict the token with the larger signal in $[\mathtt{t_c}]$ as the output (see Figure 7 in Appendix B.2 for a pictorial illustration). We formalize our result in the following theorem, and defer the proof to Appendix B.4.

**Theorem 1** (Chain of continuous thought solves graph reachability). *Fix $T_{\max} > 0$. Assume the length of the input sequence (including the continuous thoughts and the special answer token <A>) does not exceed $T_{\max}$. There exists a construction of a two-layer transformer parameters $\theta$ and the readout matrix $\mathbf{W_O} \in \mathbb{R}^{|\mathsf{Voc}| \times d}$ (where $d = O(|\mathsf{Voc}|)$) that are independent of graphs, such that for any directed graph $\mathcal{G} = (\mathcal{V}, \mathcal{E})$ with at most $n_{\max}$ nodes ($n_{\max} < |\mathsf{Voc}|$), a root node $\mathtt{r}$, two candidate destination nodes $\mathtt{c}_1, \mathtt{c}_2$, for any $C$ exceeds the diameter of the graph, it holds that*

$$\widetilde{\mathsf{TF}}_{\theta, C, \mathbf{W_O}}(\mathbf{h}_{[t_0]}) = c_{i*}.$$

### 4.4 Discussions

**Role of buffer spaces.** The buffer spaces are subspaces of the embedding to store useful information. For clarity, our construction separates the "content" and the two "buffer" spaces into different dimensions. In practice, they can be jointly projected into a more compact and lower-dimensional space. For example, we can construct $\mathbf{u} = \sum_{i=1}^{k} \mathbf{R}^{(i)}\mathbf{u}^{(i)} \in \mathbb{R}^d$ where the column space of each $\mathbf{R}^{(i)} \in \mathbb{R}^{d \times d}$ forms a subspace. Different subspaces can be (almost) orthogonal, and for each subspace, the column vectors of $\mathbf{R}^{(i)}$ can also be almost orthonormal.

**Weights of different nodes in the superposition.** In our construction, each superposition state maintains nodes with uniform weights. In practice, the weights of different nodes can vary due to factors such as training signals or the model's internal heuristic on which nodes are more likely to reach the final answer [Cohen et al., 2025]. In Section 5, we show that in practice, the training signal could bias the superposition states towards the *frontier nodes* that can be reached with exactly $i$ steps and the *optimal nodes* that can lead to the destination node.

## 5 Experiments

In this section, we conduct extensive experiments to validate our theoretical results that COCONUT outperforms discrete CoT even with many fewer layers (Section 5.2), which is indeed due to superposition states encoded in continuous thoughts during reasoning (Section 5.3).

### 5.1 Training Setup

**Model.** We adopt a GPT-2 style decoder with two transformer layers, $d_{\text{model}}=768$, $n_{\text{heads}}=8$. We train from scratch using AdamW ($\beta_1=0.9$, $\beta_2=0.95$, weight-decay $10^{-2}$) and a constant learning rate of $1 \times 10^{-4}$.

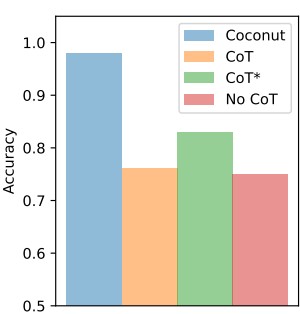

**Dataset.** We construct a subset of ProsQA [Hao et al., 2024], with questions whose solutions require 3–4 reasoning hops. Each node in the graph is injected as a dedicated token into the vocabulary. The split statistics are summarised in Table 4.

**Method.** Following Hao et al. [2024], we adopt a multi-stage training strategy with the supervision from chain-of-thoughts data. Stage $i$ teaches the model to use $i$ continuous thoughts before predicting the $i$-th node in the given chain of thought as the next token. If the index of the training stage is larger than the CoT solution length $l$, the model is trained to output the final prediction after $l$ continuous thoughts and `<A>`. We train the model for 25 epochs at each stage, and the whole training lasts 300 epochs in total. At each stage, we mix the data from the previous stage randomly with 0.1 probability, which helps improve performance in our early experiments.

Figure 4: The overall accuracy of COCONUT, CoT, CoT*(12 layers, $n_{\text{heads}} = 12$) and No CoT.

### 5.2 Overall Results

Extending the findings of Hao et al. [2024], we further show that a 2-layer transformer trained from scratch can effectively solve ProsQA when using COCONUT. As shown in Figure 4, COCONUT achieves near-perfect accuracy, while both CoT and No CoT baselines solve only about 75% of the tasks (random guessing = 50%). Even with a larger model of 12 layers and $n_{\text{heads}} = 12$, marked with ∗ in the figure, CoT improves to 83% accuracy but still cannot solve the tasks reliably.

### 5.3 Visualising Latent Reasoning

We inspect both the *attention pattern* and the *representation* of the continuous thought learned by the model to validate our theoretical construction.

**Layer 1 attention.** According to our theoretical construction, the most important function of LAYER 1 attention heads is to *copy* the source and target node tokens of an edge onto the corresponding edge token $\langle e \rangle$. Figure 5 shows a representative attention map, confirming that the model has instantiated this copying mechanism in practice.

**Layer 2 attention.** LAYER 2 is responsible for *node expansion*: each continuous thought attends to all outgoing edges from nodes that are currently reachable. To quantify this behavior, we compute, when generating $i$-th continuous thought, the aggregated attention score received by each edge token triplet (`s`, `t`, `<e>`) across all heads. 4 kinds of edges exist: (1) *Reachable*: their source node is in the

reachable set at step $i$; (2) *Not Reachable*: source node *not* in the reachable set; (3) *Frontier*: a subset of reachable edges whose source node is on the current search frontier, i.e., exactly $i$ steps away from the root node; (4) *Optimal*: a subset of frontier edges that lead to the optimal reasoning chain.

Table 1 reports group-wise means and standard deviations averaged on the test set. The model sharply concentrates its attention on *Reachable* edges, as predicted by our theoretical construction. Interestingly, there is an additional bias toward the *Frontier* subset. One possibility is that the training signal encourages the model to predict a frontier node at each step, coupled with the decaying effects for previously explored nodes. We also find that the optimal edges receive more attention scores, likely due to the supervision of multi-stage training from the CoT solution.

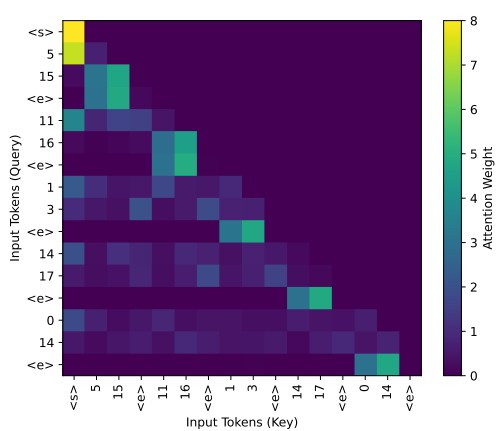

**Representation of continuous thoughts.** To verify that the continuous thought serves as superposition states for the search, we compute the inner product between the continuous thought at step $i$, $[\texttt{t}_\texttt{i}]$, and each node embedding $\mathbf{u}_v$. Similar to edge classification above, we classify nodes into *Reachable*, *Not Reachable*, *Frontier*, and *Optimal*. Figure 6 (top) plots the distribution segregated by the reasoning step $i$. As predicted, nodes within $i$ hops exhibit markedly

Figure 5: A representative example of Layer 1 attention map: the edge token `<e>` ($y$-axis) places nearly all its attention mass on its source and target nodes ($x$-axis), in line with the theoretical construction.

higher similarity scores than distant nodes. Moreover, *Frontier* nodes are noticeably closer to $[\texttt{t}_\texttt{i}]$ than other reachable nodes, illustrating that the superposition emphasizes the candidate expansion fronts. Besides, the *Optimal* nodes are even closer to $[\texttt{t}_\texttt{i}]$, also likely due to the training data always presenting the optimal path.

Collectively, these analyses confirm that the intended *superpositional search* exists in trained models: Layer 1 establishes the query context, Layer 2 expands the frontier, and the latent vectors encode a soft, parallel representation of reachable state sets, realizing the theoretical construction in Section 4. We also show in Appendix C.2 that this search pattern is consistent across multiple experiments with random seeds.

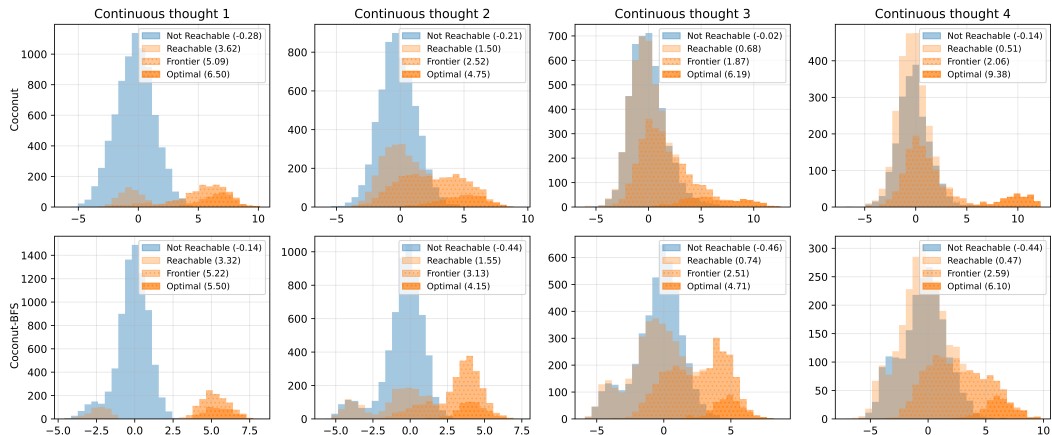

Figure 6: The histogram of inner product between the $i$-th continuous thoughts and the node embeddings. The mean value for each group is shown in the legend. Note that *Frontier* is a subset of *Reachable* nodes, and *Optimal* is a subset of *Frontier* nodes.

## 5.4 Exploration Priority

An interesting fact revealed by the visualizations in Section 5.3 is that the model allocates disproportionate attention to *optimal* edges and nodes, reminiscent of a prioritized search strategy. One hypothesis is that this behavior is an artifact of our multi-stage curriculum, which explicitly guides the model on the CoT solution at every step. To understand the effect of this multi-stage guidance, we introduce an alternative supervision method COCONUT-BFS: at stage $i$, the supervision token is drawn uniformly at random from the frontier nodes exactly $i$ hops from the root, rather than the $i$-th node in the CoT solution. All other hyperparameters are unchanged.

Table 1: Layer 2 attention scores to different edge groups at step $i$ (mean $\pm$ standard deviation).

|  | Step 1 | Step 2 | Step 3 | Step 4 |
| --- | --- | --- | --- | --- |
| Not Reachable | $0.04_{\pm0.07}$ | $0.03_{\pm0.09}$ | $0.08_{\pm0.17}$ | $0.12_{\pm0.20}$ |
| Reachable | $2.12_{\pm1.07}$ | $0.71_{\pm0.92}$ | $0.38_{\pm0.72}$ | $0.29_{\pm0.66}$ |
| –Frontier | $2.12_{\pm1.07}$ | $1.00_{\pm0.96}$ | $0.67_{\pm0.87}$ | $0.61_{\pm0.95}$ |
| –Optimal | $2.54_{\pm1.03}$ | $1.72_{\pm1.13}$ | $1.67_{\pm1.20}$ | $2.23_{\pm1.35}$ |

Our experiment results indicate that COCONUT-BFS also achieves near-perfect accuracy on ProsQA, matching the original COCONUT. Figure 6 compares the inner product distributions of the latent thoughts for both models. Remarkably, the two supervision methods converge to similar exploration strategies, even though there is no explicit guidance towards optimal nodes for COCONUT-BFS. Conversely, the original COCONUT, although trained exclusively on optimal nodes during intermediate stages, still assigns elevated weight to non-optimal frontier nodes compared to other non-frontier nodes, implicitly performing a breadth-first expansion before honing in on the solution. We leave explaining this behavior from the perspective of training dynamics as future work.

## 6 Conclusions

In this paper, we study how the chain-of-continuous-thought boosts LLM's reasoning capability by focusing on the graph reachability problem. We provided a theoretical construction of a two-layer transformer that can efficiently solve graph reachability for an $n$-vertex $D$-diameter directed graph by $D$ steps of continuous thoughts, while the best existing result on constant-depth transformers with discrete CoT requires $O(n^2)$ steps. Our construction reveals that the superposition states that encode multiple search traces simultaneously are the key to the strong reasoning capability of COCONUT, and we conducted thorough experiments to validate that our theoretical construction matches the solutions obtained by training dynamics. Several interesting future directions include: (1) Deriving a lower bound on the number of discrete CoT steps in the graph reachability problem to show a strict separation of expressivity between CoT and COCONUT; (2) Theoretical understanding of the emergence of exploration behavior with only deterministic search trace demonstration via training dynamics; (3) Advantages of reasoning in a continuous space in more general settings.

## Acknowledgements

This work was partially supported by a gift from Open Philanthropy to the Center for Human-Compatible AI (CHAI) at UC Berkeley and by NSF Grants IIS-1901252 and CCF-2211209.

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

# A Notation Details

We provide detailed descriptions of different tokens in Table 2, and the position index of different tokens or continuous thoughts in Table 3.

| Tokens | Meanings |
|---|---|
| `` | a special token denoting the beginning of the sentence |
| $s_i$ | the source node of edge $i$ |
| $t_i$ | the target node of edge $i$ |
| `<e>` | a special token marking the end of an edge |
| `<Q>` | a special token followed by two candidate nodes |
| $c_1, c_2$ | two candidate destination nodes |
| `<R>` | a special token marking the start of reasoning |
| $r$ | the root node |
| $[t_i]$ | the $i$-th continuous thought (represented by a $d$-dimensional vector) |
| `<A>` | a special token driving the model to make the final prediction |

Table 2: Meaning of each token.

| Notations | Position indices |
|---|---|
| $\mathsf{Idx}(\text{})$ | $1$ |
| $\mathsf{Idx}(s_i)$ | $3i - 1$ |
| $\mathsf{Idx}(t_i)$ | $3i$ |
| $\mathsf{Idx}(\text{<e>}, i)$ | $3i + 1$ |
| $\mathsf{Idx}(\text{<Q>})$ | $3m + 2$ |
| $\mathsf{Idx}(c_1)$ | $3m + 3$ |
| $\mathsf{Idx}(c_2)$ | $3m + 4$ |
| $\mathsf{Idx}(\text{<R>})$ | $3m + 5$ |
| $\mathsf{Idx}(r)$ | $3m + 6 = t_0$ |
| $\mathsf{Idx}([t_i])$ | $t_0 + i$ |
| $\mathsf{Idx}(\text{<A>})$ | $t_0 + C + 1 = T$ |

Table 3: Position indices of different tokens or continuous thoughts in the input sequence.

# B Missing Proofs

## B.1 Formal version and proof of Lemma 1

**Lemma 3** (Attention chooser, formal version of Lemma 1)**.** *Fix any token* `<x>` $\in$ Voc*, integer* $\ell \geq 0$*, and* $\varepsilon \in (0, 1)$*. Under sinusoidal positional encoding as defined in Definition 1, there exists a construction of* $\mathbf{K}, \mathbf{Q} \in \mathbb{R}^{(2d_{\mathsf{PE}}) \times d}$*, such that for any input sequence* $(\mathbf{h}_1, \ldots, \mathbf{h}_T)$ *that satisfies*

$$\mathbf{h}_i = \sum_{v \in \mathsf{Voc}} \lambda_{i,v} \mathbf{u}_v, \text{ where } \lambda_{i,v} \geq 0 \quad \forall v \in \mathsf{Voc}, \sum_{v \in \mathsf{Voc}} \lambda_{i,v}^2 = 1, \quad \forall i \in [T], \tag{2}$$

*and satisfies* $\langle \mathbf{u}_{\text{<x>}}, \mathbf{h}_i \rangle \in \{0, 1\}$ *(i.e., each input embedding is either equal to the embedding of token* `<x>` *or orthogonal to it) and* $\langle \mathbf{u}_{\text{<x>}}, \mathbf{h}_i \rangle = 0$ *for* $i \leq \ell$ *(i.e., the first* $\ell$ *tokens are not* `<x>`*), it holds that for any* $i \in [T]$*,*

$$\text{if } \langle \mathbf{h}_i, \mathbf{u}_{\text{<x>}} \rangle = 1, \text{ then } s_{i,i-l} > 1 - \varepsilon, \text{ otherwise } s_{i,1} > 1 - \varepsilon,$$

*where* $s_{i,j}$ *is the attention score from the* $i$*-th token to the* $j$*-th token as defined in Algorithm 2 with the input sequence* $(\mathbf{h}_1 + \mathbf{p}_1, \ldots, \mathbf{h}_T + \mathbf{p}_T)$*.*

*Proof.* Note that (2) implies that each input embedding is a normalized superposition of token embeddings in the vocabulary. We aim to construct an attention head such that when the $i$-th token is

<x>, it will pay almost all attention to the position $i - \ell$, otherwise it will pay almost all attention to the BOS token  (known as the attention sink). Define vector $\tilde{\mathbf{u}}_{\texttt{<x>}} = \sum_{v \in \mathsf{Voc}\backslash\{\texttt{<x>}\}} \tilde{\mathbf{u}}_v \in \mathbb{R}^{d_{\mathsf{TE}}}$, which is the superposition of all token embeddings in the vocabulary except for <x>. We define

$$\mathbf{Q} = \begin{bmatrix} \mathbf{0}_{d_{\mathsf{PE}} \times d_{\mathsf{TE}}} & \mathbf{0}_{d_{\mathsf{PE}} \times d_{\mathsf{TE}}} & \mathbf{0}_{d_{\mathsf{PE}} \times d_{\mathsf{TE}}} & \mathbf{I}_{d_{\mathsf{PE}}} \\ \xi \bar{\mathbf{p}}_1 \otimes \tilde{\mathbf{u}}_{\texttt{<x>}} & \mathbf{0}_{d_{\mathsf{PE}} \times d_{\mathsf{TE}}} & \mathbf{0}_{d_{\mathsf{PE}} \times d_{\mathsf{TE}}} & \mathbf{0}_{d_{\mathsf{PE}} \times d_{\mathsf{PE}}} \end{bmatrix} \in \mathbb{R}^{(2d_{\mathsf{PE}}) \times d},$$

$$\mathbf{K} = \begin{bmatrix} \mathbf{0}_{d_{\mathsf{PE}} \times d_{\mathsf{TE}}} & \mathbf{0}_{d_{\mathsf{PE}} \times d_{\mathsf{TE}}} & \mathbf{0}_{d_{\mathsf{PE}} \times d_{\mathsf{TE}}} & \eta \mathbf{R}^{(\ell)} \\ \mathbf{0}_{d_{\mathsf{PE}} \times d_{\mathsf{TE}}} & \mathbf{0}_{d_{\mathsf{PE}} \times d_{\mathsf{TE}}} & \mathbf{0}_{d_{\mathsf{PE}} \times d_{\mathsf{TE}}} & \eta \mathbf{I}_{d_{\mathsf{PE}}} \end{bmatrix} \in \mathbb{R}^{(2d_{\mathsf{PE}}) \times d},$$

where $\xi, \eta > 0$ will be specified later and $\mathbf{R}^{(\ell)}$ is defined as in Lemma 4. Therefore,

$$\mathbf{q}_i = \mathbf{Q}(\mathbf{h}_i + \mathbf{p}_i) = \begin{bmatrix} \bar{\mathbf{p}}_i \\ \xi \langle \tilde{\mathbf{u}}_{\texttt{<x>}}, \tilde{\mathbf{h}}_i \rangle \bar{\mathbf{p}}_1 \end{bmatrix}, \quad \mathbf{k}_i = \mathbf{K}(\mathbf{h}_i + \mathbf{p}_i) = \begin{bmatrix} \eta \mathbf{R}^{(\ell)} \bar{\mathbf{p}}_i \\ \eta \bar{\mathbf{p}}_i \end{bmatrix} = \begin{bmatrix} \eta \bar{\mathbf{p}}_{i+\ell} \\ \eta \bar{\mathbf{p}}_i \end{bmatrix}.$$

Now for any $1 \leq j \leq i \leq T$, we have

$$\langle \mathbf{q}_i, \mathbf{k}_j \rangle = \eta \left( \langle \bar{\mathbf{p}}_i, \bar{\mathbf{p}}_{j+\ell} \rangle + \xi \langle \tilde{\mathbf{u}}_{\texttt{<x>}}, \tilde{\mathbf{h}}_i \rangle \langle \bar{\mathbf{p}}_1, \bar{\mathbf{p}}_j \rangle \right).$$

Now we fix $i \in [T]$. We first consider the case where $\langle \mathbf{h}_i, \mathbf{u}_{\texttt{<x>}} \rangle = 1$ (which also implies $i > \ell$). By (2) and the assumption that token embeddings are orthonormal, we have

$$\langle \tilde{\mathbf{h}}_i, \tilde{\mathbf{u}}_v \rangle = \langle \mathbf{h}_i, \mathbf{u}_v \rangle = 0, \ \forall v \in \mathsf{Voc}\backslash\{\texttt{<x>}\} \implies \langle \tilde{\mathbf{h}}_i, \tilde{\mathbf{u}}_{\texttt{<x>}} \rangle = 0.$$

Therefore, we have $\langle \mathbf{q}_i, \mathbf{k}_j \rangle = \eta \langle \bar{\mathbf{p}}_i, \bar{\mathbf{p}}_{j+\ell} \rangle$. By Lemma 5, we have

$$\langle \mathbf{q}_i, \mathbf{k}_{i-\ell} \rangle = \eta \langle \bar{\mathbf{p}}_i, \bar{\mathbf{p}}_{(i-\ell)+\ell} \rangle = \eta \langle \bar{\mathbf{p}}_i, \bar{\mathbf{p}}_i \rangle = \eta d_{\mathsf{PE}}/2$$

and

$$\langle \mathbf{q}_i, \mathbf{k}_j \rangle = \eta \langle \bar{\mathbf{p}}_i, \bar{\mathbf{p}}_{j+\ell} \rangle \leq \eta d_{\mathsf{PE}}/2 - \eta \varepsilon_T, \ \forall j \neq i - \ell.$$

where $\varepsilon_T > 0$ is defined in Lemma 5. This implies $\langle \mathbf{q}_i, \mathbf{k}_j \rangle$ is maximized when $j = i - \ell$ with a non-zero gap $\eta \varepsilon_T$, and therefore, we have

$$s_{i,i-\ell} = \frac{\exp(\langle \mathbf{q}_i, \mathbf{k}_{i-\ell} \rangle)}{\sum_{j \in [i]} \exp(\langle \mathbf{q}_i, \mathbf{k}_j \rangle)} \geq \frac{\exp(\eta \varepsilon_T)}{\exp(\eta \varepsilon_T) + (i-1)}. \tag{3}$$

Now we consider the case where $\langle \mathbf{h}_i, \mathbf{u}_{\texttt{<x>}} \rangle = 0$. Again, by (2) and the orthonormal assumption of token embeddings, we have

$$\langle \tilde{\mathbf{h}}_i, \tilde{\mathbf{u}}_{\texttt{<x>}} \rangle = \sum_{v \in \mathsf{Voc}\backslash\{\texttt{<x>}\}} \langle \mathbf{h}_i, \mathbf{u}_v \rangle = \sum_{v \in \mathsf{Voc}\backslash\{\texttt{<x>}\}} \lambda_{i,v} \geq \sum_{v \in \mathsf{Voc}\backslash\{\texttt{<x>}\}} \lambda_{i,v}^2 = 1.$$

By Lemma 5, we can define $\xi = \max_{j=2,\ldots,T} \frac{2d_{\mathsf{PE}}}{\langle \bar{\mathbf{p}}_1, \bar{\mathbf{p}}_1 \rangle - \langle \bar{\mathbf{p}}_1, \bar{\mathbf{p}}_j \rangle}$, and thus

$$\xi(\langle \bar{\mathbf{p}}_1, \bar{\mathbf{p}}_1 \rangle - \langle \bar{\mathbf{p}}_1, \bar{\mathbf{p}}_j \rangle) \geq 2d_{\mathsf{PE}}, \quad \forall j \in \{2,\ldots,T\}.$$

Note that for any $j \in \{2,\ldots,T\}$, we have

$$\begin{aligned}
&\langle \mathbf{q}_i, \mathbf{k}_1 \rangle - \langle \mathbf{q}_i, \mathbf{k}_j \rangle \\
=&\eta \left( \xi \langle \tilde{\mathbf{u}}_{\texttt{<x>}}, \tilde{\mathbf{h}}_i \rangle \left( \langle \bar{\mathbf{p}}_1, \bar{\mathbf{p}}_1 \rangle - \langle \bar{\mathbf{p}}_1, \bar{\mathbf{p}}_j \rangle \right) - \left( \langle \bar{\mathbf{p}}_i, \bar{\mathbf{p}}_{j+\ell} \rangle - \langle \bar{\mathbf{p}}_i, \bar{\mathbf{p}}_{1+\ell} \rangle \right) \right) \\
\geq&\eta \left( 2d_{\mathsf{PE}} - d_{\mathsf{PE}} \right) \\
=&\eta d_{\mathsf{PE}}.
\end{aligned}$$

Therefore,

$$s_{i,1} = \frac{\exp(\langle \mathbf{q}_i, \mathbf{k}_1 \rangle)}{\sum_{j \in [i]} \exp(\langle \mathbf{q}_i, \mathbf{k}_j \rangle)} \geq \frac{\exp(\eta d_{\mathsf{PE}})}{\exp(\eta d_{\mathsf{PE}}) + (i-1)}. \tag{4}$$

By choosing a sufficiently large $\eta$, the lower bound of (3) and (4) will both exceed $1 - \varepsilon$ and thus the proof is complete. $\qquad \square$

## B.2  Proof of Lemma 2

Note that in the proof sketch of Lemma 2, we showed how to prove the lemma by induction. Here, we use an alternative way that only requires a single forward pass of the transformer. Note that the continuous thoughts are generated in an autoregressive manner, i.e., for an input sequence $\mathbf{h}_{[T]} = (\mathbf{h}_1, \ldots, \mathbf{h}_T)$, we have $\mathbf{h}_t^{(L)} = \mathsf{TF}_\theta(\mathbf{h}_{[t]})$, where $\mathbf{h}_t^{(L)}$ is defined in Algorithm 1 with the input sequence $\mathbf{h}_{[T]}$. This means for a sequence of length $t$, appending additional vectors at the end of the sequence does not affect the computation of the first $t$ positions. This means, instead of proving the following property inductively for each $c = 0, 1, \ldots, C-1$:

$$[\mathsf{t_{c+1}}] = \mathsf{TF}_\theta(\mathbf{h}_{[t_0]}, [\mathsf{t_1}], \ldots, [\mathsf{t_c}])$$

where for each $c$, it holds $[\mathsf{t_c}] = \frac{1}{\sqrt{|\mathcal{V}_c|}} \sum_{v \in \mathcal{V}_c} \mathbf{u}_v$ and $\mathbf{h}_{[t_0]}$ is defined as in Section 3, we can instead prove by a single forward pass by setting the input embedding

$$\mathbf{h}_{t_0+c} = \frac{1}{\sqrt{|\mathcal{V}_c|}} \sum_{v \in \mathcal{V}_c} \mathbf{u}_v, \quad \forall c \in [C], \tag{5}$$

and additionally setting $\mathbf{h}_T = \mathbf{h}_{t_0+C+1} = \mathbf{u}_{\texttt{<A>}}$, and prove the hidden embedding in the last layer with input $\mathbf{h}_{[T]}$ satisfies

$$\mathbf{h}_{t_0+c}^{(L)} = \frac{1}{\sqrt{|\mathcal{V}_{c+1}|}} \sum_{v \in \mathcal{V}_{c+1}} \mathbf{u}_v, \quad \forall 0 \le c \le C.$$

In Appendix B.3, we construct the parameter for each component of the two-layer transformer. Finally, Proposition B.4 precisely provides the result we desire.

## B.3  Construction of transformer parameters

**Proposition B.1** (First layer attention). *For any fixed $\varepsilon \in (0, 1/2)$, there exists a construction of key and query matrices for first layer attention heads $\{\mathbf{Q}^{(0,h)}, \mathbf{K}^{(0,h)}\}_{h=0,\ldots,4}$ s.t. for any input embeddings $\mathbf{h} = (\mathbf{h}_1, \ldots, \mathbf{h}_t)$ satisfying the format specified in Appendix B.2, the value of following terms exceed $1 - \varepsilon$ for all $i \in [t]$:*

- *$s_{i,i-2}^{(0,0)}$ if $\mathbf{h}_i = \mathbf{u}_{\texttt{<e>}}$, and $s_{i,1}^{(0,0)}$ otherwise; $s_{i,i-1}^{(0,1)}$ if $\mathbf{h}_i = \mathbf{u}_{\texttt{<e>}}$, and $s_{i,1}^{(0,1)}$ otherwise;*

- *$s_{i,i-2}^{(0,2)}$ if $\mathbf{h}_i = \mathbf{u}_{\texttt{<R>}}$, and $s_{i,1}^{(0,2)}$ otherwise; $s_{i,i-1}^{(0,3)}$ if $\mathbf{h}_i = \mathbf{u}_{\texttt{<R>}}$, and $s_{i,1}^{(3)}$ otherwise;*

- *$s_{i,i-1}^{(0,4)}$ if $\mathbf{h}_i = \mathbf{u}_{\texttt{<A>}}$, and $s_{i,1}^{(0,4)}$ otherwise;*

*where $s_i^{(0,h)} \leftarrow \mathsf{SoftMax}(\langle \mathbf{q}_i^{(h)}, \mathbf{k}_1^{(h)} \rangle, \ldots, \langle \mathbf{q}_i^{(h)}, \mathbf{k}_i^{(h)} \rangle) \in \mathbb{R}^i$ and $\mathbf{k}_i^{(h)} = \mathbf{K}^{(0,h)} \mathbf{h}_i^{(0)}, \mathbf{q}_i^{(h)} = \mathbf{Q}^{(0,h)} \mathbf{h}_i^{(0)}, \mathbf{h}_i^{(0)} = \mathbf{h}_i + \mathbf{p}_i$.*

*Proof.* Note that each of the attention heads is an attention chooser as defined in Lemma 3. Therefore, the proposition directly holds by constructing each attention head as described in Lemma 3.

$\square$

By Proposition B.1, each edge token `<e>` will pay attention to its corresponding source node and target node. Also, the reasoning token `<R>` will pay attention to two candidate destination nodes, and the answer token `<A>` will pay attention to the last continuous thought. When no token or position needs to be paid attention to for some attention heads, it will dump attention to the BOS token ``, a phenomenon known as attention sink. Since the attention after softmax cannot exactly be zero, there will be undesired tokens with noise-level magnitude copied to each position. The subsequent MLP layer will filter out the noise. We formalize the above statements rigorously in the following proposition.

**Proposition B.2** (First layer MLP). *When the input sequence $\mathbf{h}_{[T]}$ satisfies conditions in Appendix B.2, there exists a construction of $\theta_{\mathsf{Attn}}^{(0,h)}$ for $h \in \{0, \ldots, 4\}$, $\theta_{\mathsf{MLP}}^{(0)}$ such that the output of the first layer $\mathbf{h}^{(1)}$ satisfies:*

- $\mathbf{h}^{(1)}_{\mathsf{Idx}(\texttt{<e>},i)} = \frac{1}{\sqrt{3}}[\tilde{\mathbf{u}}_{\texttt{<e>}}^\top \ \tilde{\mathbf{u}}_{s_i}^\top \ \tilde{\mathbf{u}}_{t_i}^\top \ \mathbf{0}_{d_{\mathsf{PE}}}^\top]^\top, \quad \forall i \in [m]$

- $\mathbf{h}^{(1)}_{\mathsf{Idx}(\texttt{<R>})} = \frac{1}{\sqrt{3}}[\tilde{\mathbf{u}}_{\texttt{<R>}}^\top \ \mathbf{0}_{d_{\mathsf{TE}}}^\top \ (\tilde{\mathbf{u}}_{c_1} + \tilde{\mathbf{u}}_{c_2})^\top \ \mathbf{0}_{d_{\mathsf{PE}}}^\top]^\top$

- $\mathbf{h}^{(1)}_{\mathsf{Idx}(\texttt{<A>})} = \frac{1}{\sqrt{|\mathcal{V}_C|+1}} \left[\tilde{\mathbf{u}}_{\texttt{<A>}}^\top \ \sum_{v \in \mathcal{V}_C} \tilde{\mathbf{u}}_v^\top \ \mathbf{0}_{d_{\mathsf{TE}}}^\top \ \mathbf{0}_{d_{\mathsf{PE}}}^\top \right]^\top$

- $\mathbf{h}^{(1)}_{\mathsf{Idx}(\texttt{[}t_i\texttt{]})} = \frac{1}{\sqrt{|\mathcal{V}_i|}} \left[\sum_{v \in \mathcal{V}_i} \tilde{\mathbf{u}}_v^\top \ \mathbf{0}_{d_{\mathsf{TE}}}^\top \ \mathbf{0}_{d_{\mathsf{TE}}}^\top \ \mathbf{0}_{d_{\mathsf{PE}}}^\top \right]^\top, \forall i \in [C]$

- $\mathbf{h}^{(1)}_i = \mathbf{h}_i$ *for other* $i \in [T]$.

*Proof.* We use the construction in Proposition B.1 for $\{\mathbf{Q}^{(0,h)}, \mathbf{K}^{(0,h)}\}_{h=0,\dots,4}$. For each position, after attending to the desired tokens, each attention head will copy the attended tokens to one of the two buffer spaces. Formally, we provide the construction of value and output matrices for each head below. First, the value matrices are constructed as

$$\mathbf{V}^{(0,h)} = [\mathbf{I}_{d_{\mathsf{TE}}} - \tilde{\mathbf{u}}_{\texttt{}} \otimes \tilde{\mathbf{u}}_{\texttt{}} \quad \mathbf{0}_{d_{\mathsf{TE}} \times (d-d_{\mathsf{TE}})}] \in \mathbb{R}^{d_{\mathsf{TE}} \times d}, \quad h = 0, \dots, 4.$$

By construction, we have $\mathbf{v}_i^{(h)} = \mathbf{V}^{(0,h)} \mathbf{h}_i^{(0)} = \text{content}(\mathbf{h}_i) \cdot \mathbb{1}\{i > 1\} = \tilde{\mathbf{h}}_i \cdot \mathbb{1}\{i > 1\}$ for $i \in [T], h \in \{0, \dots, 4\}$. This is due to the input format specified in Appendix B.2, which ensures that only $\tilde{\mathbf{h}}_1$ contains $\tilde{\mathbf{u}}_{\texttt{}}$.

Now we construct output matrices such that the $h$-th attention head copies the content of the attended token to buffer 1 for $h = 0, 4$, and to buffer 2 for $h = 1, 2, 3$. Formally, we construct

$$\mathbf{O}^{(0,h)} = [\mathbf{0}_{d_{\mathsf{TE}} \times d_{\mathsf{TE}}} \quad \mathbf{I}_{d_{\mathsf{TE}}} \quad \mathbf{0}_{d_{\mathsf{TE}} \times d_{\mathsf{TE}}} \quad \mathbf{0}_{d_{\mathsf{TE}} \times d_{\mathsf{PE}}}]^\top \in \mathbb{R}^{d \times d_{\mathsf{TE}}}, \quad h = 0, 4,$$

$$\mathbf{O}^{(0,h)} = [\mathbf{0}_{d_{\mathsf{TE}} \times d_{\mathsf{TE}}} \quad \mathbf{0}_{d_{\mathsf{TE}} \times d_{\mathsf{TE}}} \quad \mathbf{I}_{d_{\mathsf{TE}}} \quad \mathbf{0}_{d_{\mathsf{TE}} \times d_{\mathsf{PE}}}]^\top \in \mathbb{R}^{d \times d_{\mathsf{TE}}}, \quad h = 1, 2, 3.$$

Therefore, it holds that $\mathbf{O}^{(0,h)} \mathbf{v}_i^{(h)} = [\mathbf{0}_{d_{\mathsf{TE}}}^\top \ \tilde{\mathbf{h}}_i^\top \cdot \mathbb{1}\{i > 1\} \ \mathbf{0}_{d_{\mathsf{TE}}}^\top \ \mathbf{0}_{d_{\mathsf{PE}}}^\top]^\top$ for $h = 0, 4$ and $\mathbf{O}^{(0,h)} \mathbf{v}_i^{(h)} = [\mathbf{0}_{d_{\mathsf{TE}}}^\top \ \mathbf{0}_{d_{\mathsf{TE}}}^\top \ \tilde{\mathbf{h}}_i^\top \cdot \mathbb{1}\{i > 1\} \ \mathbf{0}_{d_{\mathsf{PE}}}^\top]^\top$ for $h = 1, 2, 3$, and thus we have

$$\mathsf{Attn}_{\theta_{\mathsf{Attn}}^{(0,h)}}(\mathbf{h}^{(0)})_i = \sum_{j=1}^{i} s_{i,j}^{(0,h)} \mathbf{O}^{(0,h)} \mathbf{v}_j^{(h)} = \left[\mathbf{0}_{d_{\mathsf{TE}}}^\top \ \sum_{j=2}^{i} s_{i,j}^{(0,h)} \tilde{\mathbf{h}}_j^\top \ \mathbf{0}_{d_{\mathsf{TE}}}^\top \ \mathbf{0}_{d_{\mathsf{PE}}}^\top \right]^\top, \quad h = 0, 4,$$

$$\mathsf{Attn}_{\theta_{\mathsf{Attn}}^{(0,h)}}(\mathbf{h}^{(0)})_i = \sum_{j=1}^{i} s_{i,j}^{(0,h)} \mathbf{O}^{(0,h)} \mathbf{v}_j^{(h)} = \left[\mathbf{0}_{d_{\mathsf{TE}}}^\top \ \mathbf{0}_{d_{\mathsf{TE}}}^\top \ \sum_{j=2}^{i} s_{i,j}^{(0,h)} \tilde{\mathbf{h}}_j^\top \ \mathbf{0}_{d_{\mathsf{PE}}}^\top \right]^\top, \quad h = 1, 2, 3,$$

where $s_{i,j}^{(0,h)}$ is defined as in Proposition B.1. Therefore, the output at position $i$ of the first attention layer is

$$\mathbf{h}_i^{(0.5)} = \mathbf{h}_i^{(0)} + \sum_{h=0}^{4} \mathsf{Attn}_{\theta_{\mathsf{Attn}}^{(0,h)}}(\mathbf{h}^{(0)})_i = \left[\tilde{\mathbf{h}}_i^\top \ \sum_{h=0,4}\sum_{j=2}^{i} s_{i,j}^{(0,h)} \tilde{\mathbf{h}}_j^\top \ \sum_{h=1}^{3}\sum_{j=2}^{i} s_{i,j}^{(0,h)} \tilde{\mathbf{h}}_j^\top \ \mathsf{pos}(\mathbf{p}_i)^\top \right]^\top.$$

Next, we construct the parameters of the MLP of the first layer. For notation convenience and by the input format in Appendix B.2, we can decompose $\tilde{\mathbf{h}}_i = \sum_{j=1}^{V} \lambda_{i,j}^{(0)} \tilde{\mathbf{u}}_j = \mathbf{U} \boldsymbol{\lambda}_i^{(0)}$ where $\boldsymbol{\lambda}_i^{(0)} = [\lambda_{i,1}^{(0)}, \lambda_{i,2}^{(0)}, \dots, \lambda_{i,V}^{(0)}]^\top \in \mathbb{R}^V$. Let

$$\mathsf{MLP}_{\theta_{\mathsf{MLP}}^{(0)}}(\mathbf{h}^{(0.5)})_i = \mathbf{W}_2^{(0)} \sigma_\varepsilon^{(0)}(\mathbf{W}_1^{(0)} \mathbf{h}_i^{(0.5)}) \in \mathbb{R}^d$$

which is a two-layer neural network. Let $\mathbf{W}_1^{(0)} = [\mathrm{diag}(\mathbf{U}^\top, \mathbf{U}^\top, \mathbf{U}^\top), \mathbf{0}_{(3V) \times d_{\mathsf{PE}}}] \in \mathbb{R}^{3V \times d}$. Then

$$\mathbf{W}_1^{(0)} \mathbf{h}_i^{(0.5)} = \begin{bmatrix} \mathbf{U}^\top \mathbf{U} \boldsymbol{\lambda}_i^{(0)} \\ \mathbf{U}^\top \mathbf{U} \sum_{h=0,4}\sum_{j=2}^{i} s_{i,j}^{(0,h)} \boldsymbol{\lambda}_j^{(0)} \\ \mathbf{U}^\top \mathbf{U} \sum_{h=1}^{3}\sum_{j=2}^{i} s_{i,j}^{(0,h)} \boldsymbol{\lambda}_j^{(0)} \end{bmatrix} = \begin{bmatrix} \boldsymbol{\lambda}_i^{(0)} \\ \sum_{h=0,4}\sum_{j=2}^{i} s_{i,j}^{(0,h)} \boldsymbol{\lambda}_j^{(0)} \\ \sum_{h=1}^{3}\sum_{j=2}^{i} s_{i,j}^{(0,h)} \boldsymbol{\lambda}_j^{(0)} \end{bmatrix} \in \mathbb{R}^{3V}.$$

Let $\sigma_\varepsilon^{(0)}(\cdot)$ be a coordinate-wise non-linearity such that $\sigma_\varepsilon^{(0)}(x) = \mathbb{1}\{x \geq \varepsilon\}$ for $x \in \mathbb{R}$. We choose $\varepsilon = \frac{1}{2\sqrt{n}}$ where $n$ is the (maximum possible) number of vertices in the graph. We denote $i_*^{(h)} = \arg\max_{j \leq i} s_{i,j}^{(0,h)}$, i.e., $i_*^{(h)}$ is the position that position $i$ pays the most attention within head $h$. By Proposition B.1, we can construct key and query matrices such that $s_{i,i_*^{(h)}}^{(0,h)} > 1 - \varepsilon/5$ for any $i$ and $h$. Also, by the input format especially by (5), $\lambda_{i,k}^{(0)} \in \{0,1\} \cup \{1/\sqrt{i} \mid i \in [n]\}$, which implies that any non-zero $\lambda_{i,k}^{(0)}$ satisfies $\lambda_{i,k}^{(0)} \in [2\varepsilon, 1]$ and all non-zero entries of $\boldsymbol{\lambda}_i^{(0)}$ share the same value for any fixed $i$. Then by definition of $\sigma_\varepsilon^{(0)}(\cdot)$, we have $\sigma_\varepsilon^{(0)}(\boldsymbol{\lambda}_i^{(0)}) = \frac{\boldsymbol{\lambda}_i^{(0)}}{\|\boldsymbol{\lambda}_i^{(0)}\|_\infty}$.

Now we analyze $\sum_{j=2}^{i} s_{i,j}^{(0,h)} \boldsymbol{\lambda}_j^{(0)}$. For any $k \in [V]$, we consider $\sum_{j=2}^{i} s_{i,j}^{(0,h)} \lambda_{j,k}^{(0)}$. If $i_*^{(h)} = 1$, then we have

$$\sum_{j=2}^{i} s_{i,j}^{(0,h)} \lambda_{j,k}^{(0)} \leq \left( \sum_{j=2}^{i} s_{i,j}^{(0,h)} \right) \cdot \max_{2 \leq j \leq i} \lambda_{j,k}^{(0)} < \frac{\varepsilon}{5} \cdot 1 = \frac{\varepsilon}{5}.$$

If $i_*^{(h)} > 1$, we have

$$\sum_{j=2}^{i} s_{i,j}^{(0,h)} \lambda_{j,k}^{(0)} = s_{i,i_*^{(h)}}^{(0,h)} \lambda_{i_*^{(h)},k}^{(0)} + \sum_{j=2, j \neq i_*^{(h)}}^{i} s_{i,j}^{(0,h)} \lambda_{j,k}^{(0)}.$$

When $\lambda_{i_*^{(h)},k}^{(0)} = 0$, we can obtain that

$$\sum_{j=2}^{i} s_{i,j}^{(0,h)} \lambda_{j,k}^{(0)} = \sum_{j=2, j \neq i_*^{(h)}}^{i} s_{i,j}^{(0,h)} \lambda_{j,k}^{(0)} \leq \left( \sum_{j=2, j \neq i_*^{(h)}}^{i} s_{i,j}^{(0,h)} \right) \cdot \max_{2 \leq j \leq i} \lambda_{j,k}^{(0)} < \frac{\varepsilon}{5}.$$

When $\lambda_{i_*^{(h)},k}^{(0)} > 0$, we can obtain that

$$\sum_{j=2}^{i} s_{i,j}^{(0,h)} \lambda_{j,k}^{(0)} \geq s_{i,i_*^{(h)}}^{(0,h)} \lambda_{i_*^{(h)},k}^{(0)} \geq (1 - \varepsilon/5) \cdot 2\varepsilon \geq \varepsilon.$$

Therefore, $\sigma_\varepsilon^{(0)}(\sum_{h=0,4} \sum_{j=2}^{i} s_{i,j}^{(0,h)} \lambda_{j,k}^{(0)}) = \mathbb{1}\left\{ \bigcup_{h=0,4} \left( \left(i_*^{(h)} > 1\right) \cap \left(\lambda_{i_*^{(h)},k}^{(0)} > 0\right) \right) \right\}, \forall k \in [V]$. Also, by Proposition B.1, for any $i \in [T]$ and $k \in [V]$, there is at most one $h \in \{0, \ldots, 4\}$ such that $i_*^{(h)} > 1$ and $\lambda_{i_*^{(h)},k}^{(0)} > 0$ hold simultaneously. Therefore, $\sigma_\varepsilon^{(0)}(\sum_{h=0,4} \sum_{j=2}^{i} s_{i,j}^{(0,h)} \lambda_{j,k}^{(0)}) = \sum_{h=0,4} \mathbb{1}\{i_*^{(h)} > 1\} \cdot \mathbb{1}\{\lambda_{i_*^{(h)},k}^{(0)} > 0\}, \forall k \in [V]$, and thus we can write this compactly

$$\sigma_\varepsilon^{(0)}\left( \sum_{h=0,4} \sum_{j=2}^{i} s_{i,j}^{(0,h)} \boldsymbol{\lambda}_j^{(0)} \right) = \sum_{h=0,4} \mathbb{1}\{i_*^{(h)} > 1\} \cdot \mathbb{1}\{\boldsymbol{\lambda}_{i_*^{(h)}}^{(0)} > \mathbf{0}_V\}.$$

Similarly, we have

$$\sigma_\varepsilon^{(0)}\left( \sum_{h=1}^{3} \sum_{j=2}^{i} s_{i,j}^{(0,h)} \boldsymbol{\lambda}_j^{(0)} \right) = \sum_{h=1}^{3} \mathbb{1}\{i_*^{(h)} > 1\} \cdot \mathbb{1}\{\boldsymbol{\lambda}_{i_*^{(h)}}^{(0)} > \mathbf{0}_V\}.$$

Therefore,

$$\sigma_\varepsilon^{(0)}(\mathbf{W}_1^{(0)} \mathbf{h}_i^{(0.5)}) = \begin{bmatrix} \boldsymbol{\lambda}_i^{(0)} / \|\boldsymbol{\lambda}_i^{(0)}\|_\infty \\ \sum_{h=0,4} \mathbb{1}\{i_*^{(h)} > 1\} \cdot \mathbb{1}\{\boldsymbol{\lambda}_{i_*^{(h)}}^{(0)} > \mathbf{0}_V\} \\ \sum_{h=1}^{3} \mathbb{1}\{i_*^{(h)} > 1\} \cdot \mathbb{1}\{\boldsymbol{\lambda}_{i_*^{(h)}}^{(0)} > \mathbf{0}_V\} \end{bmatrix} \in \mathbb{R}^{3V}.$$

Finally, we set $\mathbf{W}_2^{(0)} = \mathbf{W}_1^{(0)\top} = [\mathrm{diag}(\mathbf{U}^\top, \mathbf{U}^\top, \mathbf{U}^\top), \mathbf{0}_{(3V) \times d_{\mathsf{PE}}}]^\top \in \mathbb{R}^{d \times 3V}$, and thus

$$
\mathsf{MLP}_{\theta_{\mathsf{MLP}}^{(0)}}(\mathbf{h}^{(0.5)})_i = \begin{bmatrix} \mathbf{U}\boldsymbol{\lambda}_i^{(0)}/\|\boldsymbol{\lambda}_i^{(0)}\|_\infty \\ \mathbf{U}\sum_{h=0,4} \mathbb{1}\{i_*^{(h)} > 1\} \cdot \mathbb{1}\{\boldsymbol{\lambda}_{i_*^{(h)}}^{(0)} > \mathbf{0}_V\} \\ \mathbf{U}\sum_{h=1}^3 \mathbb{1}\{i_*^{(h)} > 1\} \cdot \mathbb{1}\{\boldsymbol{\lambda}_{i_*^{(h)}}^{(0)} > \mathbf{0}_V\} \\ \mathbf{0}_{d_{\mathsf{PE}}} \end{bmatrix} \in \mathbb{R}^d.
$$

Now we derive the output of the MLP for different positions $i$.

For $i = \mathsf{Idx}(\texttt{<e>}, k) = 3k + 1$ where $k \in [m]$, we have $i_*^{(0)} = i - 2 = \mathsf{Idx}(\mathbf{s}_k)$ and $i_*^{(1)} = i - 1 = \mathsf{Idx}(\mathbf{t}_k)$. Note that $i_*^{(h)} = 1$ for $h = 2, 3, 4$. Also, $\boldsymbol{\lambda}_{\mathsf{Idx}(\mathbf{s}_k)}^{(0)} = \boldsymbol{e}_{\mathbf{s}_k}$, $\boldsymbol{\lambda}_{\mathsf{Idx}(\mathbf{t}_k)}^{(0)} = \boldsymbol{e}_{\mathbf{t}_k}$ and $\boldsymbol{\lambda}_i^{(0)} = \boldsymbol{e}_{\texttt{<e>}}$ are all $V$-dimensional one-hot vectors. Therefore, we have

$$
\mathsf{MLP}_{\theta_{\mathsf{MLP}}^{(0)}}(\mathbf{h}^{(0.5)})_{\mathsf{Idx}(\texttt{<e>},k)} = [\tilde{\mathbf{u}}_{\texttt{<e>}}^\top \ \tilde{\mathbf{u}}_{\mathbf{s}_k}^\top \ \tilde{\mathbf{u}}_{\mathbf{t}_k}^\top \ \mathbf{0}_{d_{\mathsf{PE}}}^\top]^\top.
$$

For $i = \mathsf{Idx}(\texttt{<R>}) = 3m + 5$, we have $i_*^{(2)} = i - 2 = \mathsf{Idx}(\mathbf{c}_1)$ and $i_*^{(3)} = i - 1 = \mathsf{Idx}(\mathbf{c}_2)$. Note that $i_*^{(h)} = 1$ for $h = 0, 1, 4$. Also, $\boldsymbol{\lambda}_{i-2}^{(0)} = \boldsymbol{e}_{\mathbf{c}_1}$ and $\boldsymbol{\lambda}_{i-1}^{(0)} = \boldsymbol{e}_{\mathbf{c}_2}$, $\boldsymbol{\lambda}_i^{(0)} = \boldsymbol{e}_{\texttt{<R>}}$. Therefore,

$$
\mathsf{MLP}_{\theta_{\mathsf{MLP}}^{(0)}}(\mathbf{h}^{(0.5)})_{\mathsf{Idx}(\texttt{<R>})} = [\tilde{\mathbf{u}}_{\texttt{<R>}}^\top \ \mathbf{0}_{d_{\mathsf{TE}}}^\top \ (\tilde{\mathbf{u}}_{\mathbf{c}_1} + \tilde{\mathbf{u}}_{\mathbf{c}_2})^\top \ \mathbf{0}_{d_{\mathsf{PE}}}^\top]^\top.
$$

For $i = \mathsf{Idx}(\texttt{<A>}) = t_0 + C + 1$, we have $i_*^{(4)} = i - 1$ and $i_*^{(h)} = 1$ for $0 \le h \le 3$. Note that $\boldsymbol{\lambda}_i^{(0)} = \boldsymbol{e}_{\texttt{<A>}}$, $\boldsymbol{\lambda}_{i-1}^{(0)} = \boldsymbol{\lambda}_{\mathsf{Idx}(\texttt{[t}_C\texttt{]})}^{(0)} = \frac{1}{\sqrt{|\mathcal{V}_C|}} \sum_{v \in \mathcal{V}_C} \boldsymbol{e}_v$ where $\mathcal{V}_C$ is the set of vertices reachable from $\mathbf{r}$ within $C$ steps. Then we have

$$
\mathsf{MLP}_{\theta_{\mathsf{MLP}}^{(0)}}(\mathbf{h}^{(0.5)})_{\mathsf{Idx}(\texttt{<A>})} = \left[ \tilde{\mathbf{u}}_{\texttt{<A>}}^\top \ \sum_{v \in \mathcal{V}_C} \tilde{\mathbf{u}}_v^\top \ \mathbf{0}_{d_{\mathsf{TE}}}^\top \ \mathbf{0}_{d_{\mathsf{PE}}}^\top \right]^\top.
$$

For $i = \mathsf{Idx}(\texttt{[t}_c\texttt{]}) = t_0 + c$ for $c \in [C]$, we have $i_*^{(h)} = 1$ for all $h$ and $\boldsymbol{\lambda}_i^{(0)} = \frac{1}{\sqrt{|\mathcal{V}_c|}} \sum_{v \in \mathcal{V}_c} \boldsymbol{e}_v$, and thus

$$
\mathsf{MLP}_{\theta_{\mathsf{MLP}}^{(0)}}(\mathbf{h}^{(0.5)})_{\mathsf{Idx}(\texttt{[t}_c\texttt{]})} = \left[ \sum_{v \in \mathcal{V}_c} \tilde{\mathbf{u}}_v^\top \ \mathbf{0}_{d_{\mathsf{TE}}}^\top \ \mathbf{0}_{d_{\mathsf{TE}}}^\top \ \mathbf{0}_{d_{\mathsf{PE}}}^\top \right]^\top.
$$

For remaining $i$, we have $i_*^{(h)} = 1$ for all $h$ and $\boldsymbol{\lambda}_i^{(0)}$ is one-hot. So

$$
\mathsf{MLP}_{\theta_{\mathsf{MLP}}^{(0)}}(\mathbf{h}^{(0.5)})_i = \left[ \tilde{\mathbf{h}}_i^\top \ \mathbf{0}_{d_{\mathsf{TE}}}^\top \ \mathbf{0}_{d_{\mathsf{TE}}}^\top \ \mathbf{0}_{d_{\mathsf{PE}}}^\top \right]^\top = \mathbf{h}_i.
$$

By applying layer normalization, we can obtain the final result. $\qquad \square$

Note that Proposition B.2 shows that after the first layer, each $\texttt{<e>}$ token will copy its corresponding source and target token embeddings into its two buffer spaces, respectively. For the second layer, since it is the last layer, we only need to focus on positions for current thoughts $\mathsf{Idx}(\texttt{[t}_c\texttt{]}) = t_0 + c$ and the position for the final prediction $\mathsf{Idx}(\texttt{<A>}) = T$. Since we only need one attention head for the second attention layer, we will omit the index for heads and only keep the index for layers.

**Proposition B.3** (Second layer attention). *Under the construction of Proposition B.2 and input format specified in Appendix B.2, for any fixed $\varepsilon \in (0, 1/2)$, there exists a construction of the second-layer attention parameters $\theta_{\mathsf{Attn}}^{(1)} = (\mathbf{Q}^{(1)}, \mathbf{K}^{(1)}, \mathbf{V}^{(1)}, \mathbf{O}^{(1)})$ s.t.*

$$
\mathbf{h}_{\mathsf{Idx}(\texttt{[t}_c\texttt{]})}^{(1.5)} = \begin{bmatrix} \frac{1}{\sqrt{|\mathcal{V}_c|}} \sum_{v \in \mathcal{V}_c} \tilde{\mathbf{u}}_v + \frac{\sum_{j=1, \mathbf{s}_j \in \mathcal{V}_c}^m \tilde{\mathbf{u}}_{\mathbf{t}_j}}{\sqrt{3}|\{j \mid \mathbf{s}_j \in \mathcal{V}_c, j \in [m]\}|} \\ \mathbf{0}_{d - d_{\mathsf{TE}}} \end{bmatrix} + \begin{bmatrix} \mathbf{U}\boldsymbol{\varepsilon}^{(c)} \\ \mathbf{0}_{d - d_{\mathsf{TE}}} \end{bmatrix}, \ \forall c \in \{0\} \cup [C],
$$

*and*

$$
\mathbf{h}_{\mathsf{Idx}(\texttt{<A>})}^{(1.5)} = \begin{bmatrix} \frac{1}{\sqrt{|\mathcal{V}_C|+1}} \tilde{\mathbf{u}}_{\texttt{<A>}} + \frac{1}{\sqrt{3}}(\tilde{\mathbf{u}}_{\mathbf{c}_1} + \tilde{\mathbf{u}}_{\mathbf{c}_2}) \\ \frac{1}{\sqrt{|\mathcal{V}_C|+1}} \sum_{v \in \mathcal{V}_C} \tilde{\mathbf{u}}_v \\ \mathbf{0}_{d_{\mathsf{TE}}+d_{\mathsf{PE}}} \end{bmatrix} + \begin{bmatrix} \mathbf{U}\boldsymbol{\varepsilon}^{(C+1)} \\ \mathbf{0}_{d_{\mathsf{TE}}} \\ \mathbf{0}_{d_{\mathsf{TE}}+d_{\mathsf{PE}}} \end{bmatrix},
$$

*where $\boldsymbol{\varepsilon}^{(c)} \in \mathbb{R}^V$ and $\|\boldsymbol{\varepsilon}^{(c)}\|_\infty < \varepsilon$ for all $c \in \{0\} \cup [C + 1]$.*

*Proof.* In the second attention layer, we aim to construct key and query matrices such that (1) the current thought $[\mathtt{t_c}]$ will pay attention to all edges if the source node of the edge is contained in the superposition (see Figure 3); (2) the answer token $\mathtt{<A>}$ pays large attention to the reasoning token $\mathtt{<R>}$ which stores the two candidate destination tokens in its buffer space (see Figure 7).

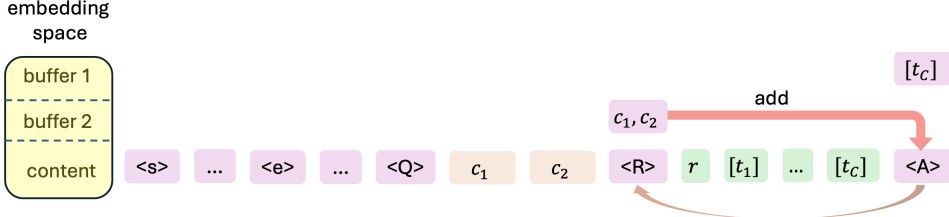

Figure 7: Illustration of the second layer attention mechanism for final prediction.

First, we construct

$$\mathbf{Q}^{(1)} = [\mathbf{I}_{d_{\mathsf{TE}}} \quad \mathbf{0}_{d_{\mathsf{TE}} \times d_{\mathsf{TE}}} \quad \mathbf{0}_{d_{\mathsf{TE}} \times d_{\mathsf{TE}}} \quad \mathbf{0}_{d_{\mathsf{TE}} \times d_{\mathsf{PE}}}] \in \mathbb{R}^{d_{\mathsf{TE}} \times d},$$

$$\mathbf{K}^{(1)} = [\tau \tilde{\mathbf{u}}_{\mathtt{<A>}} \otimes \tilde{\mathbf{u}}_{\mathtt{<R>}} \quad \tau \mathbf{I}_{d_{\mathsf{TE}}} \quad \mathbf{0}_{d_{\mathsf{TE}} \times d_{\mathsf{TE}}} \quad \mathbf{0}_{d_{\mathsf{TE}} \times d_{\mathsf{PE}}}] \in \mathbb{R}^{d_{\mathsf{TE}} \times d},$$

where the value of $\tau > 0$ will be decided later.

We define $\mathbf{q}_i = \mathbf{Q}^{(1)} \mathbf{h}_i^{(1)}$, $\mathbf{k}_i = \mathbf{K}^{(1)} \mathbf{h}_i^{(1)}$ and let

$$s_{i,j} = \frac{\exp(\langle \mathbf{q}_i, \mathbf{k}_j \rangle)}{\sum_{j' \leq i} \exp(\langle \mathbf{q}_i, \mathbf{k}_{j'} \rangle)}.$$

Note that we have $T = \mathsf{Idx}(\mathtt{<A>}) = t_0 + C + 1$. By the construction of the query and key matrices, we have $\mathbf{q}_i = \mathsf{content}(\mathbf{h}_i^{(1)}), \forall i \in [T]$, $\mathbf{k}_{\mathsf{Idx}(\mathtt{<R>})} = \tau \cdot \mathsf{buffer}_1(\mathbf{h}_{\mathsf{Idx}(\mathtt{<R>})}^{(1)}) + \tau \tilde{\mathbf{u}}_{\mathtt{<A>}}$ and $\mathbf{k}_i = \tau \cdot \mathsf{buffer}_1(\mathbf{h}_i^{(1)})$ for other $i$. Now we consider attention weights for $i = \mathsf{Idx}([\mathtt{t_c}]) = t_0 + c, \forall c \in \{0\} \cup [C]$ and $i = \mathsf{Idx}(\mathtt{<A>}) = T$.

For $i = \mathsf{Idx}([\mathtt{t_c}])$ for any fixed $c \in \{0\} \cup [C]$, we have $\mathbf{q}_i = \frac{1}{\sqrt{|\mathcal{V}_c|}} \sum_{v \in \mathcal{V}_c} \tilde{\mathbf{u}}_v$. By Proposition B.2, we have $\mathbf{k}_{\mathsf{Idx}(\mathtt{<e>},j)} = \frac{\tau}{\sqrt{3}} \tilde{\mathbf{u}}_{\mathbf{s}_j}$ for $j \in [m]$, $\mathbf{k}_{\mathsf{Idx}(\mathtt{<R>})} = \tau \tilde{\mathbf{u}}_{\mathtt{<A>}}$ and $\mathbf{k}_j = 0$ for other $j \leq i$. Define $\mathcal{I}_c = \{\mathsf{Idx}(\mathtt{<e>}, j) \mid \mathbf{s}_j \in \mathcal{V}_c \text{ for } j \in [m]\}$. Therefore,

$$\langle \mathbf{q}_i, \mathbf{k}_j \rangle = \frac{\tau}{\sqrt{3|\mathcal{V}_c|}} \mathbb{1}\{j \in \mathcal{I}_c\}.$$

Then we have

$$s_{i,j} = \frac{\exp\left(\tau/\sqrt{3|\mathcal{V}_c|}\right) \cdot \mathbb{1}\{j \in \mathcal{I}_c\}}{|\mathcal{I}_c| \exp\left(\tau/\sqrt{3|\mathcal{V}_c|}\right) + (i - |\mathcal{I}_c|)} + \frac{\mathbb{1}\{j \notin \mathcal{I}_c\}}{|\mathcal{I}_c| \exp\left(\tau/\sqrt{3|\mathcal{V}_c|}\right) + (i - |\mathcal{I}_c|)}.$$

For $i = \mathsf{Idx}(\mathtt{<A>}) = T$, note that $\mathbf{q}_T = \frac{1}{\sqrt{|\mathcal{V}_C|+1}} \tilde{\mathbf{u}}_{\mathtt{<A>}}$. Then $\langle \mathbf{q}_T, \mathbf{k}_j \rangle$ is non-zero only when $j = \mathsf{Idx}(\mathtt{<R>})$, and the inner product is $\frac{\tau}{\sqrt{|\mathcal{V}_C|+1}}$. Then we have

$$s_{T,j} = \frac{\exp\left(\tau/\sqrt{|\mathcal{V}_C|+1}\right) \cdot \mathbb{1}\{j = \mathsf{Idx}(\mathtt{<R>})\}}{\exp\left(\tau/\sqrt{|\mathcal{V}_C|+1}\right) + (T-1)} + \frac{\mathbb{1}\{j \neq \mathsf{Idx}(\mathtt{<R>})\}}{\exp\left(\tau/\sqrt{|\mathcal{V}_C|+1}\right) + (T-1)}.$$

By choosing a large enough $\tau$, we have that

$$s_{\mathsf{Idx}([\mathtt{t_c}]),j} = \left(\frac{1}{|\mathcal{I}_c|} - \varepsilon_{c,1}\right) \cdot \mathbb{1}\{j \in \mathcal{I}_c\} + \varepsilon_{c,2} \cdot \mathbb{1}\{j \notin \mathcal{I}_c\}, \quad \forall c \in \{0\} \cup [C],$$

$$s_{T,j} = (1 - \varepsilon_{C+1,1}) \cdot \mathbb{1}\{j = \mathsf{Idx}(\mathtt{<R>})\}\} + \varepsilon_{C+1,2} \cdot \mathbb{1}\{j \neq \mathsf{Idx}(\mathtt{<R>})\},$$

where $\varepsilon_{c,1}, \varepsilon_{c,2} \in (0, \varepsilon/T), \forall c \in \{0\} \cup [C+1]$.

Now we construct the value and output matrix where

$$\mathbf{V}^{(1)} = [\mathbf{0}_{d_{\text{TE}} \times d_{\text{TE}}} \quad \mathbf{0}_{d_{\text{TE}} \times d_{\text{TE}}} \quad \mathbf{I}_{d_{\text{TE}}} \quad \mathbf{0}_{d_{\text{TE}} \times d_{\text{PE}}}] \in \mathbb{R}^{d_{\text{TE}} \times d},$$

$$\mathbf{O}^{(1)} = [\mathbf{I}_{d_{\text{TE}}} \quad \mathbf{0}_{d_{\text{TE}} \times (d - d_{\text{TE}})}]^\top \in \mathbb{R}^{d \times d_{\text{TE}}}.$$

Then $\mathbf{v}_i \triangleq \mathbf{V}^{(1)} \mathbf{h}_i^{(1)} = \mathsf{buffer}_2(\mathbf{h}_i^{(1)}) \in \mathbb{R}^{d_{\text{TE}}}$ reads from buffer space 2, and $\mathbf{O}^{(1)} \mathbf{v}_i = [\mathsf{buffer}_2(\mathbf{h}_i^{(1)})^\top \quad \mathbf{0}_{d-d_{\text{TE}}}^\top]^\top \in \mathbb{R}^d$ writes to the content space. Note that

$$\mathsf{Attn}_{\theta_{\text{Attn}}^{(1)}}(\mathbf{h}^{(1)})_i = \sum_{j=1}^{i} s_{i,j} \mathbf{O}^{(1)} \mathbf{v}_j = \begin{bmatrix} \sum_{j=1}^{i} s_{i,j} \mathsf{buffer}_2(\mathbf{h}_j^{(1)}) \\ \mathbf{0}_{d-d_{\text{TE}}} \end{bmatrix}.$$

Since, $\mathbf{h}_i^{(1.5)} = \mathbf{h}_i^{(1)} + \mathsf{Attn}_{\theta_{\text{Attn}}^{(1)}}(\mathbf{h}^{(1)})_i$, we have

$$\mathbf{h}_{\mathsf{Idx}([\mathbf{t}_c])}^{(1.5)} = \begin{bmatrix} \frac{1}{\sqrt{|\mathcal{V}_c|}} \sum_{v \in \mathcal{V}_c} \tilde{\mathbf{u}}_v + (\frac{1}{|\mathcal{I}_c|} - \varepsilon_{c,1}) \sum_{j=1, \mathbf{s}_j \in \mathcal{V}_c}^{m} \frac{\tilde{\mathbf{u}}_{\mathbf{t}_j}}{\sqrt{3}} + \varepsilon_{c,2} \sum_{j \in [t_0+c] \setminus \mathcal{I}_c} \mathsf{buffer}_2(\mathbf{h}_j^{(1)}) \\ \mathbf{0}_{d-d_{\text{TE}}} \end{bmatrix}$$

and

$$\mathbf{h}_T^{(1.5)} = \begin{bmatrix} \frac{1}{\sqrt{|\mathcal{V}_C|+1}} \tilde{\mathbf{u}}_{\texttt{<A>}} + (1 - \varepsilon_{C+1,1}) \frac{\tilde{\mathbf{u}}_{\mathbf{c}_1} + \tilde{\mathbf{u}}_{\mathbf{c}_2}}{\sqrt{3}} + \varepsilon_{C+1,2} \sum_{j \in [T] \setminus \{\mathsf{Idx}(\texttt{<R>})\}} \mathsf{buffer}_2(\mathbf{h}_j^{(1)}) \\ \frac{1}{\sqrt{|\mathcal{V}_C|+1}} \sum_{v \in \mathcal{V}_C} \tilde{\mathbf{u}}_v \\ \mathbf{0}_{d_{\text{TE}}+d_{\text{PE}}} \end{bmatrix}.$$

Combining the fact that $\mathsf{buffer}_2(\mathbf{h}_j^{(1)})$ is either zero or equal to the superposition of $\tilde{\mathbf{u}}_v$ for some $v \in \mathsf{Voc}$ with norm less than 1, we can obtain the final result. $\qquad\square$

After the second-layer attention, the current thought $\mathbf{h}_{t_0+c}^{(1.5)}$ now contains all vertices in $\mathcal{V}_c$ and all vertices that can be reached within one step from $\mathcal{V}_c$, which is $\mathcal{V}_{c+1}$ by definition. The subsequent MLP layer will then equalize the weights of each vertex and eliminate the noise in the continuous thought.

**Proposition B.4** (Second layer MLP). *Under the construction of Proposition B.3 and the input format specified by Appendix B.2, there exists a construction of $\theta_{\text{MLP}}^{(1)}$ such that the output of the second-layer MLP satisfies:*

- $\mathbf{h}_{\mathsf{Idx}([\mathbf{t}_c])}^{(2)} = \frac{1}{\sqrt{|\mathcal{V}_{c+1}|}} \sum_{v \in \mathcal{V}_{c+1}} \mathbf{u}_v, \quad \forall c \in \{0\} \cup [C],$

- $\mathbf{h}_{\mathsf{Idx}(\texttt{<A>})}^{(2)} = \frac{1}{\sqrt{|\mathcal{V}_C|+5}} (\mathbf{u}_{\mathbf{c}_1} + \mathbf{u}_{\mathbf{c}_2} + \mathbf{u}_{\texttt{<A>}} + \sum_{v \in \mathcal{V}_C} \mathbf{u}_v).$

*Proof.* Similar to Proposition B.2, we let

$$\mathsf{MLP}_{\theta_{\text{MLP}}^{(1)}}(\mathbf{h}^{(1.5)})_i = \mathbf{W}_2^{(1)} \sigma_\varepsilon^{(1)}(\mathbf{W}_1^{(1)} \mathbf{h}_i^{(1.5)}) \in \mathbb{R}^d$$

where $\varepsilon > 0$ and the (elementwise) non-linearity $\sigma_\varepsilon^{(1)}(\cdot)$ will be specified later. We only consider $\mathbf{h}_i^{(1.5)}$ where $i = t_0 + c$ for $0 \le c \le C+1$. By Proposition B.3, we can decompose

$$\mathbf{h}_{t_0+c}^{(1.5)} = \begin{bmatrix} \mathbf{U}\boldsymbol{\lambda}^{(c)} \\ \mathbf{0}_{d_{\text{TE}}} \\ \mathbf{0}_{d_{\text{TE}}+d_{\text{PE}}} \end{bmatrix} \forall c \in \{0\} \cup [C], \quad \mathbf{h}_T^{(1.5)} = \begin{bmatrix} \mathbf{U}\boldsymbol{\eta}^{(1)} \\ \mathbf{U}\boldsymbol{\eta}^{(2)} \\ \mathbf{0}_{d_{\text{TE}}+d_{\text{PE}}} \end{bmatrix},$$

where $\boldsymbol{\lambda}^{(c)} = [\lambda_1^{(c)}, \dots, \lambda_V^{(c)}]^\top, \boldsymbol{\eta}^{(1)} = [\eta_1^{(1)}, \dots \eta_V^{(1)}]^\top, \boldsymbol{\eta}^{(2)} = [\eta_1^{(2)}, \dots \eta_V^{(2)}]^\top \in \mathbb{R}^V$. Let

$$\mathbf{W}_1^{(1)} = \begin{bmatrix} \mathbf{U}^\top & & \mathbf{0}_{V \times (d_{\text{TE}}+d_{\text{PE}})} \\ & \mathbf{U}^\top & \mathbf{0}_{V \times (d_{\text{TE}}+d_{\text{PE}})} \end{bmatrix} \in \mathbb{R}^{(2V) \times d}, \ \mathbf{W}_2^{(1)} = \begin{bmatrix} \mathbf{U} & \mathbf{U} \\ \mathbf{0}_{(d-d_{\text{TE}}) \times V} & \mathbf{0}_{(d-d_{\text{TE}}) \times V} \end{bmatrix} \in \mathbb{R}^{d \times (2V)},$$

and $\sigma_\varepsilon^{(1)}(x) = \mathbb{1}\{x \geq \varepsilon\}$. Then

$$\mathsf{MLP}_{\theta_{\mathsf{MLP}}^{(1)}}(\mathbf{h}^{(1.5)})_{t_0+c} = \mathbf{W}_2^{(1)}\sigma_\varepsilon^{(1)}(\mathbf{W}_1^{(1)}\mathbf{h}_{t_0+c}^{(1.5)})$$

$$= \mathbf{W}_2^{(1)}\sigma_\varepsilon^{(1)}\left(\begin{bmatrix} \mathbf{U}^\top & & \mathbf{0}_{V\times(d_{\mathsf{TE}}+d_{\mathsf{PE}})} \\ & \mathbf{U}^\top & \mathbf{0}_{V\times(d_{\mathsf{TE}}+d_{\mathsf{PE}})} \end{bmatrix}\begin{bmatrix} \mathbf{U}\boldsymbol{\lambda}^{(c)} \\ \mathbf{0}_{d_{\mathsf{TE}}} \\ \mathbf{0}_{d_{\mathsf{TE}}+d_{\mathsf{PE}}} \end{bmatrix}\right)$$

$$= \begin{bmatrix} \mathbf{U} & \mathbf{U} \\ \mathbf{0}_{(d-d_{\mathsf{TE}})\times V} & \mathbf{0}_{(d-d_{\mathsf{TE}})\times V} \end{bmatrix}\sigma_\varepsilon^{(1)}\left(\begin{bmatrix} \mathbf{U}^\top\mathbf{U}\boldsymbol{\lambda}^{(c)} \\ \mathbf{0}_V \end{bmatrix}\right)$$

$$= \begin{bmatrix} \mathbf{U}\sigma_\varepsilon^{(1)}(\boldsymbol{\lambda}^{(c)}) \\ \mathbf{0}_{d-d_{\mathsf{TE}}} \end{bmatrix},$$

and similarly,

$$\mathsf{MLP}_{\theta_{\mathsf{MLP}}^{(1)}}(\mathbf{h}^{(1.5)})_T = \mathbf{W}_2^{(1)}\sigma_\varepsilon^{(1)}(\mathbf{W}_1^{(1)}\mathbf{h}_T^{(1.5)})$$

$$= \mathbf{W}_2^{(1)}\sigma_\varepsilon^{(1)}\left(\begin{bmatrix} \mathbf{U}^\top & & \mathbf{0}_{V\times(d_{\mathsf{TE}}+d_{\mathsf{PE}})} \\ & \mathbf{U}^\top & \mathbf{0}_{V\times(d_{\mathsf{TE}}+d_{\mathsf{PE}})} \end{bmatrix}\begin{bmatrix} \mathbf{U}\boldsymbol{\eta}^{(1)} \\ \mathbf{U}\boldsymbol{\eta}^{(2)} \\ \mathbf{0}_{d_{\mathsf{TE}}+d_{\mathsf{PE}}} \end{bmatrix}\right)$$

$$= \begin{bmatrix} \mathbf{U} & \mathbf{U} \\ \mathbf{0}_{(d-d_{\mathsf{TE}})\times V} & \mathbf{0}_{(d-d_{\mathsf{TE}})\times V} \end{bmatrix}\sigma_\varepsilon^{(1)}\left(\begin{bmatrix} \mathbf{U}^\top\mathbf{U}\boldsymbol{\eta}^{(1)} \\ \mathbf{U}^\top\mathbf{U}\boldsymbol{\eta}^{(2)} \end{bmatrix}\right)$$

$$= \begin{bmatrix} \mathbf{U}\left(\sigma_\varepsilon^{(1)}(\boldsymbol{\eta}^{(1)}) + \sigma_\varepsilon^{(1)}(\boldsymbol{\eta}^{(2)})\right) \\ \mathbf{0}_{d-d_{\mathsf{TE}}} \end{bmatrix}.$$

Now we choose $\varepsilon = \frac{1}{4n}$ where $n = |\mathcal{V}|$ is the number of vertices in the graph. By Proposition B.3, we can make sure that $\|\varepsilon^{(c)}\|_\infty < \frac{1}{16n}$ for all $c \in \{0\} \cup [C+1]$ where $\varepsilon^{(c)}$ is defined in Proposition B.3. We define $\Delta_c = \{\mathbf{t}_j \mid \mathbf{s}_j \in \mathcal{V}_c, j \in [m]\}$ which is the set of vertices that can be reached within one step from the currently explored vertex set $\mathcal{V}_c$. By definition, we have $\mathcal{V}_{c+1} = \mathcal{V}_c \cup \Delta_c$. We consider any fixed $c \in \{0\} \cup [C]$. For $v \in \mathcal{V}_c \cup \Delta_c$, it holds that

$$\lambda_v^{(c)} \geq \min\{1/\sqrt{|\mathcal{V}_c|}, 1/(\sqrt{3}|\Delta_c|)\} + \varepsilon_v^{(c)} \geq \frac{1}{2n} - \frac{1}{16n} > \frac{1}{4n} = \varepsilon.$$

For other $v$,

$$\lambda_v^{(c)} \leq \varepsilon_v^{(c)} < \frac{1}{16n} < \varepsilon.$$

Therefore, $\sigma_\varepsilon^{(1)}(\lambda_v^{(c)}) = \mathbb{1}\{v \in \mathcal{V}_c \cup \Delta_c\} = \mathbb{1}\{v \in \mathcal{V}_{c+1}\}$, and thus we have

$$\mathsf{MLP}_{\theta_{\mathsf{MLP}}^{(1)}}(\mathbf{h}^{(1.5)})_{t_0+c} = \begin{bmatrix} \sum_{v\in\mathcal{V}_{c+1}} \tilde{\mathbf{u}}_v \\ \mathbf{0}_{d-d_{\mathsf{TE}}} \end{bmatrix}, \quad \forall c \in \{0\} \cup [C].$$

Also, we have $\eta_{\mathsf{c}_1}^{(1)} = \frac{1}{\sqrt{3}} + \varepsilon_{\mathsf{c}_1}^{(C+1)}$, $\eta_{\mathsf{c}_2}^{(1)} = \frac{1}{\sqrt{3}} + \varepsilon_{\mathsf{c}_2}^{(C+1)}$, $\eta_{\mathsf{<A>}}^{(1)} = \frac{1}{\sqrt{|\mathcal{V}_C|+1}} + \varepsilon_{\mathsf{<A>}}^{(C+1)}$, and $\eta_v^{(1)} < \frac{1}{4n}$ for other $v \in \mathsf{Voc}$. Moreover, $\eta_v^{(2)} = \frac{1}{\sqrt{|\mathcal{V}_C|+1}} \cdot \mathbb{1}\{v \in \mathcal{V}_C\}$, which implies $\sigma_\varepsilon^{(1)}(\eta_v^{(2)}) = \mathbb{1}\{v \in \mathcal{V}_C\}$. Then

$$\mathsf{MLP}_{\theta_{\mathsf{MLP}}^{(1)}}(\mathbf{h}^{(1.5)})_T = \begin{bmatrix} \tilde{\mathbf{u}}_{\mathsf{c}_1} + \tilde{\mathbf{u}}_{\mathsf{c}_2} + \tilde{\mathbf{u}}_{\mathsf{<A>}} + \sum_{v\in\mathcal{V}_C} \tilde{\mathbf{u}}_v \\ \mathbf{0}_{d-d_{\mathsf{TE}}} \end{bmatrix}.$$

Note that by our construction of the input sequence, one and only one of $\mathsf{c}_i$ is in $\mathcal{V}_C$, and thus $\|\mathsf{MLP}_{\theta_{\mathsf{MLP}}^{(1)}}(\mathbf{h}^{(1.5)})_T\|_2 = \sqrt{|\mathcal{V}_C| + 5}$.

By applying layer normalization, we can obtain the final result. $\qquad\square$

### B.4 Proof of the main theorem

*Proof of Theorem 1.* We use the same construction of $\theta$ as in Lemma 2 where the maximum input length is $T_{\max}$. By Lemma 2, we have $\mathbf{h}_{t_0+c} = \frac{1}{\sqrt{|\mathcal{V}_c|}}\sum_{v\in\mathcal{V}_c}\mathbf{u}_v$, for $0 \leq c \leq C$. Then $(\mathbf{h}_1, \ldots, \mathbf{h}_T)$

satisfies the input format specified in Appendix B.2, and by Proposition B.4, we have

$$\mathsf{TF}_\theta(\mathbf{h}_1,\ldots,\mathbf{h}_T) = \mathbf{h}_T^{(2)} = \frac{1}{\sqrt{|\mathcal{V}_C|+5}}(\mathbf{u}_{\mathsf{c}_1} + \mathbf{u}_{\mathsf{c}_2} + \mathbf{u}_{\texttt{<A>}} + \sum_{v\in\mathcal{V}_C}\mathbf{u}_v)$$

$$= \frac{1}{\sqrt{|\mathcal{V}_C|+5}}(2\mathbf{u}_{\mathsf{c}_{i*}} + \mathbf{u}_{\mathsf{c}_{3-i*}} + \mathbf{u}_{\texttt{<A>}} + \sum_{v\in\mathcal{V}_C\setminus\{\mathsf{c}_{i*}\}}\mathbf{u}_v)$$

since $\mathsf{c}_{i*} \in \mathcal{V}_C$ and $\mathsf{c}_{3-i*} \notin \mathcal{V}_C$ by the property the graph reachability problem. We set $\mathbf{W_O} = [\mathbf{u}_1, \mathbf{u}_2, \ldots, \mathbf{u}_V]^\top$ and then

$$\mathbf{W_O}\mathsf{TF}_\theta(\mathbf{h}_1,\ldots,\mathbf{h}_T) = \frac{1}{\sqrt{|\mathcal{V}_C|+5}}(2\mathbf{e}_{\mathsf{c}_{i*}} + \mathbf{e}_{\mathsf{c}_{3-i*}} + \mathbf{e}_{\texttt{<A>}} + \sum_{v\in\mathcal{V}_C\setminus\{\mathsf{c}_{i*}\}}\mathbf{e}_v)$$

$$\implies \widetilde{\mathsf{TF}}_{\theta,C,\mathbf{W_O}}(\mathbf{h}_{[t_0]}) = \arg\max_{v\in\mathsf{Voc}}\mathbf{W_O}\mathsf{TF}_\theta(\mathbf{h}_1,\ldots,\mathbf{h}_T) = \mathsf{c}_{i*}.$$

$\square$

### B.5 Properties of sinusoidal positional encodings

**Lemma 4.** *For any integer $\ell \geq 1$, there exists a matrix $\mathbf{R}^{(\ell)} \in \mathbb{R}^{d_{\mathsf{PE}}\times d_{\mathsf{PE}}}$, such that*

$$\bar{\mathbf{p}}_{i+\ell} = \mathbf{R}^{(\ell)}\bar{\mathbf{p}}_i, \quad \forall i \geq 1.$$

*Proof.* For any $j \in [d_{\mathsf{PE}}/2]$, we construction a two-dimensional rotation matrix

$$\mathbf{R}^{(\ell,j)} = \begin{bmatrix} \cos(\ell\cdot\omega^j) & -\sin(\ell\cdot\omega^j) \\ \sin(\ell\cdot\omega^j) & \cos(\ell\cdot\omega^j) \end{bmatrix}.$$

By definition of the rotation matrix, for any $i \geq 1$, we have

$$\mathbf{R}^{(\ell,j)}\begin{bmatrix}\bar{p}_{i,2j-1} \\ \bar{p}_{i,2j}\end{bmatrix} = \mathbf{R}^{(\ell,j)}\begin{bmatrix}\cos\left(i\cdot\omega^j\right) \\ \sin\left(i\cdot\omega^j\right)\end{bmatrix} = \begin{bmatrix}\cos\left((i+\ell)\cdot\omega^j\right) \\ \sin\left((i+\ell)\cdot\omega^j\right)\end{bmatrix} = \begin{bmatrix}\bar{p}_{i+\ell,2j-1} \\ \bar{p}_{i+\ell,2j}\end{bmatrix}.$$

Then by setting $\mathbf{R}^{(\ell)} = \mathrm{diag}\{\mathbf{R}^{(\ell,1)},\mathbf{R}^{(\ell,2)},\ldots,\mathbf{R}^{(\ell,d_{\mathsf{PE}}/2)}\} \in \mathbb{R}^{d_{\mathsf{PE}}\times d_{\mathsf{PE}}}$ we can obtain the desired result. $\square$

**Lemma 5.** *For any integer $T \geq 1$, there exists $\varepsilon_T > 0$, s.t. $\langle\bar{\mathbf{p}}_i,\bar{\mathbf{p}}_j\rangle \leq d_{\mathsf{PE}}/2 - \varepsilon_T$ for any $i,j \in [T]$ where $i \neq j$. Also, $\langle\bar{\mathbf{p}}_i,\bar{\mathbf{p}}_i\rangle = d_{\mathsf{PE}}/2$ for all $i \in [T]$.*

*Proof.* For any $i,j \in [T]$, we have

$$\langle\bar{\mathbf{p}}_i,\bar{\mathbf{p}}_j\rangle = \sum_{k=1}^{d_{\mathsf{PE}}} p_{i,k}p_{j,k}$$

$$= \sum_{k=1}^{d_{\mathsf{PE}}/2}(\cos(i\cdot\omega^k)\cos(j\cdot\omega^k) + \sin(i\cdot\omega^k)\sin(j\cdot\omega^k))$$

$$= \sum_{k=1}^{d_{\mathsf{PE}}/2}\cos((i-j)\omega^k).$$

Note that for $i = j$, we can directly obtain that $\langle p_i, p_j\rangle = d_{\mathsf{PE}}/2$ since $\cos((i-j)\omega^k) = \cos 0 = 1$ for $i = j$ and any $k \in [d_{\mathsf{PE}}/2]$. Also, since $(i-j)\omega^k = \frac{i-j}{M^{2k/d_{\mathsf{PE}}}}$ is not a multiplier of $2\pi$ for $i \neq j, k \in [d_{\mathsf{PE}}/2]$, we have $\cos((i-j)\omega^k) < 1$. Let

$$\varepsilon_{T,k} = 1 - \max_{i,j\in[T],i\neq j}\cos((i-j)\omega^k) > 0, \quad \forall k \in [d_{\mathsf{PE}}/2],$$

and let $\varepsilon_T = \sum_{k=1}^{d_{\mathsf{PE}}/2}\varepsilon_{T,k} > 0$. Therefore, for $i \neq j$, we have

$$\langle\bar{\mathbf{p}}_i,\bar{\mathbf{p}}_j\rangle = \sum_{k=1}^{d_{\mathsf{PE}}/2}\cos((i-j)\omega^k) \leq \sum_{k=1}^{d_{\mathsf{PE}}/2}(1-\varepsilon_{T,k}) = d_{\mathsf{PE}}/2 - \varepsilon_T.$$

$\square$

## B.6 Implementing attention chooser under rotary position embedding

In this section, we discuss how to extend our constructions under sinusoidal positional encoding, a widely used absolute positional encoding, to the rotary position embedding (RoPE) [Su et al., 2024], a widely used relative positional encoding, to solve graph reachability. Since in our construction, the positional encoding only functions in the first attention layer, and the building blocks of the first attention layers are attention choosers, we mainly focus on how to build the attention chooser block under RoPE.

Since RoPE is a relative positional encoding, we don't use $\mathbf{p}_i$ in computation and thus don't need the last $d_{\mathsf{PE}}$ entries in the embedding vectors. Also, since the attention chooser only uses the information in the content space, we omit the two buffer spaces and only keep two dimensions to make our construction clean and assume $d = d_{\mathsf{TE}} + 2$ in this section for the simplicity of the notation. Therefore, we have $\mathbf{u}_v = [\tilde{\mathbf{u}}_v^\top, 0, 0]^\top \in \mathbb{R}^d$ for all $v \in \mathsf{Voc}$ in this section, where $\{\tilde{\mathbf{u}}_v\}_{v \in \mathsf{Voc}}$ are orthonormal. Also, assume the input embedding of attention layers satisfies $\mathbf{h}_i = [\tilde{\mathbf{h}}_i^\top, 1, 0]^\top \in \mathbb{R}^d$. This can be achieved easily by adding a bias before the attention layer or modifying the $(d_{\mathsf{TE}} + 1)$-th entry of token embeddings.

Recall the definition of RoPE below:

**Definition 2** (Rotary position embedding [Su et al., 2024]). *Let $d$ be even. For any integer $i$ and any index $k \in [d/2]$, we define*

$$\mathbf{R}^{(i,k)} = \begin{bmatrix} \cos(i \cdot \omega^k) & -\sin(i \cdot \omega^k) \\ \sin(i \cdot \omega^k) & \cos(i \cdot \omega^k) \end{bmatrix}.$$

*Let*

$$\mathbf{R}_{\mathsf{TE}}^{(i)} = \mathrm{diag}\{\mathbf{R}^{(i,1)}, \mathbf{R}^{(i,2)}, \dots, \mathbf{R}^{(i,d_{\mathsf{TE}}/2)}\} \in \mathbb{R}^{d_{\mathsf{TE}} \times d_{\mathsf{TE}}}$$

*and*

$$\mathbf{R}^{(i)} = \mathrm{diag}\{\mathbf{R}^{(i,1)}, \mathbf{R}^{(i,2)}, \dots, \mathbf{R}^{(i,d/2)}\} \in \mathbb{R}^{d \times d},$$

*where $\omega = M^{-2/d}$ and $M > 0$ is a large constant integer, e.g., $M = 10^4$ as chosen in Su et al. [2024]. We make one additional assumption that $M \geq T$ where $T$ is the maximum length of the input sequence. Then the query vector $\mathbf{q}_i$ and key vector $\mathbf{k}_i$ in Algorithm 2 will be calculated as*

$$\mathbf{q}_i = \mathbf{R}^{(i)}\mathbf{Q}\mathbf{h}_i, \quad \mathbf{k}_i = \mathbf{R}^{(i)}\mathbf{K}\mathbf{h}_i.$$

Now we show the counterpart of Lemma 3 under RoPE below.

**Lemma 6** (Attention chooser under RoPE). *Fix any token `<x>` $\in \mathsf{Voc}$, integer $\ell \geq 0$, and $\varepsilon \in (0, 1)$. Under rotary position embedding as defined in Definition 2, there exists a construction of $\mathbf{K}, \mathbf{Q} \in \mathbb{R}^{(2d) \times d}$, such that for any input sequence $(\mathbf{h}_1, \dots, \mathbf{h}_T)$ that satisfies*

$$\tilde{\mathbf{h}}_i = \sum_{v \in \mathsf{Voc}} \lambda_{i,v} \tilde{\mathbf{u}}_v, \text{ where } \lambda_{i,v} \geq 0 \quad \forall v \in \mathsf{Voc}, \sum_{v \in \mathsf{Voc}} \lambda_{i,v}^2 = 1, \quad \forall i \in [T], \tag{6}$$

*and satisfies $\langle \tilde{\mathbf{u}}_{\texttt{<x>}}, \tilde{\mathbf{h}}_i \rangle \in \{0, 1\}$ (i.e., each input embedding is either equal to the embedding of token `<x>` or orthogonal to it) and $\langle \tilde{\mathbf{u}}_{\texttt{<x>}}, \tilde{\mathbf{h}}_i \rangle = 0$ for $i \leq \ell$ (i.e., the first $\ell$ tokens are not `<x>`), it holds that for any $i \in [T]$,*

$$\text{if } \langle \tilde{\mathbf{h}}_i, \tilde{\mathbf{u}}_{\texttt{<x>}} \rangle = 1, \text{ then } s_{i,i-l} > 1 - \varepsilon, \text{ otherwise } s_{i,1} > 1 - \varepsilon,$$

*where $s_{i,j}$ is the attention score from the $i$-th token to the $j$-th token as defined in Algorithm 2 with the modification defined in Definition 2 with the input sequence $(\mathbf{h}_1, \dots, \mathbf{h}_T)$.*

*Proof.* Note that (6) implies that each input embedding is a normalized superposition of token embeddings in the vocabulary (ignoring the last two entries). We aim to construct an attention head such that when the $i$-th token is `<x>`, it will pay almost all attention to the position $i - \ell$, otherwise it will pay almost all attention to the BOS token `` (known as the attention sink). We define vector $\tilde{\mathbf{u}}_{\bar{\texttt{x}}} = \sum_{v \in \mathsf{Voc} \setminus \{\texttt{<x>}\}} \tilde{\mathbf{u}}_v \in \mathbb{R}^d$, which is the superposition of all token embeddings in the vocabulary except for `<x>`. Moreover, for any integer $i$, we define $\mathbf{p}_i^{\mathsf{TE}} = (p_{i,1}, \dots, p_{i,d_{\mathsf{TE}}})^\top \in \mathbb{R}^{d_{\mathsf{TE}}}$, and define $\mathbf{p}_i^{\widetilde{\mathsf{TE}}} = (p_{i,d-1}, p_{i,d})^\top \in \mathbb{R}^2$, where for any $j \in [d/2]$, we have

$$p_{i,2j-1} = \cos(i \cdot \omega^j), \quad p_{i,2j} = \sin(i \cdot \omega^j),$$

and $\omega = M^{-2/d_{\mathsf{PE}}}$ where $M$ is defined in Definition 2.

Now we construct query and key matrices to be

$$\mathbf{Q} = \begin{bmatrix} \mathbf{0}_{d_{\mathsf{TE}} \times d_{\mathsf{TE}}} & \mathbf{p}_0^{\mathsf{TE}} \otimes \mathbf{1}_2 \\ \xi \mathbf{p}_{-T}^{\widetilde{\mathsf{TE}}} \otimes \tilde{\mathbf{u}}_{<\bar{\mathsf{x}}>} & \mathbf{0}_{2 \times 2} \end{bmatrix} \in \mathbb{R}^{d \times d},$$

$$\mathbf{K} = \begin{bmatrix} \mathbf{0}_{d_{\mathsf{TE}} \times d_{\mathsf{TE}}} & \eta \mathbf{R}_{\mathsf{TE}}^{(\ell)}(\mathbf{p}_0^{\mathsf{TE}} \otimes \mathbf{1}_2) \\ \mathbf{0}_{2 \times d_{\mathsf{TE}}} & \eta \mathbf{p}_0^{\widetilde{\mathsf{TE}}} \otimes \mathbf{1}_2 \end{bmatrix} \in \mathbb{R}^{d \times d},$$

where $\xi, \eta > 0$ will be specified later and $\mathbf{R}_{\mathsf{TE}}^{(\ell)} \in \mathbb{R}^{d_{\mathsf{TE}} \times d_{\mathsf{TE}}}, \mathbf{R}^{(-T,d/2)} \in \mathbb{R}^{2 \times 2}$ are defined as in Definition 2 and $\mathbf{1}_m$ denotes the all-one vector of dimension $m$. By Lemma 4, one can calculate that

$$\mathbf{R}_{\mathsf{TE}}^{(j)} \mathbf{p}_i^{\mathsf{TE}} = \mathbf{p}_{i+j}^{\mathsf{TE}}, \quad \mathbf{R}^{(j,d/2)} \mathbf{p}_i^{\widetilde{\mathsf{TE}}} = \mathbf{p}_{i+j}^{\widetilde{\mathsf{TE}}}, \quad \text{for any integers } i, j.$$

Therefore,

$$\mathbf{q}_i = \mathbf{R}^{(i)} \mathbf{Q} \mathbf{h}_i = \mathbf{R}^{(i)} \begin{bmatrix} \mathbf{p}_0^{\mathsf{TE}} \\ \xi \langle \tilde{\mathbf{u}}_{<\bar{\mathsf{x}}>}, \tilde{\mathbf{h}}_i \rangle \mathbf{p}_{-T}^{\widetilde{\mathsf{TE}}} \end{bmatrix},$$

$$\mathbf{k}_i = \mathbf{R}^{(i)} \mathbf{K} \mathbf{h}_i = \mathbf{R}^{(i)} \begin{bmatrix} \eta \mathbf{R}_{\mathsf{TE}}^{(\ell)} \mathbf{p}_0^{\mathsf{TE}} \\ \eta \mathbf{p}_0^{\widetilde{\mathsf{TE}}} \end{bmatrix} = \mathbf{R}^{(i)} \begin{bmatrix} \eta \mathbf{p}_\ell^{\mathsf{TE}} \\ \eta \mathbf{p}_0^{\widetilde{\mathsf{TE}}} \end{bmatrix}.$$

Then for any $1 \leq j \leq i \leq T$, we have

$$\langle \mathbf{q}_i, \mathbf{k}_j \rangle = \eta \left( \langle \mathbf{R}_{\mathsf{TE}}^{(i)} \mathbf{p}_0^{\mathsf{TE}}, \mathbf{R}_{\mathsf{TE}}^{(j)} \mathbf{p}_\ell^{\mathsf{TE}} \rangle + \xi \langle \tilde{\mathbf{u}}_{<\bar{\mathsf{x}}>}, \tilde{\mathbf{h}}_i \rangle \langle \mathbf{R}^{(i)} \mathbf{p}_{-T}^{\widetilde{\mathsf{TE}}}, \mathbf{R}^{(j)} \mathbf{p}_0^{\widetilde{\mathsf{TE}}} \rangle \right)$$

$$= \eta \left( \langle \mathbf{p}_i^{\mathsf{TE}}, \mathbf{p}_{j+\ell}^{\mathsf{TE}} \rangle + \xi \langle \tilde{\mathbf{u}}_{<\bar{\mathsf{x}}>}, \tilde{\mathbf{h}}_i \rangle \cdot \langle \mathbf{p}_{i-j-T}^{\widetilde{\mathsf{TE}}}, \mathbf{p}_0^{\widetilde{\mathsf{TE}}} \rangle \right).$$

Now we fix $i \in [T]$. We first consider the case where $\langle \tilde{\mathbf{h}}_i, \tilde{\mathbf{u}}_{<\bar{\mathsf{x}}>} \rangle = 1$ (which also implies $i > \ell$). By (6) and the assumption that token embeddings are orthonormal, we have

$$\langle \tilde{\mathbf{h}}_i, \tilde{\mathbf{u}}_v \rangle = 0, \ \forall v \in \mathsf{Voc} \backslash \{<\mathsf{x}>\} \implies \langle \tilde{\mathbf{h}}_i, \tilde{\mathbf{u}}_{<\bar{\mathsf{x}}>} \rangle = 0.$$

Therefore, we have $\langle \mathbf{q}_i, \mathbf{k}_j \rangle = \eta \langle \mathbf{p}_i^{\mathsf{TE}}, \mathbf{p}_{j+\ell}^{\mathsf{TE}} \rangle$. By Lemma 5, we have

$$\langle \mathbf{q}_i, \mathbf{k}_{i-\ell} \rangle = \eta \langle \mathbf{p}_i^{\mathsf{TE}}, \mathbf{p}_{(i-\ell)+\ell}^{\mathsf{TE}} \rangle = \eta \langle \mathbf{p}_i^{\mathsf{TE}}, \mathbf{p}_i^{\mathsf{TE}} \rangle = \eta d_{\mathsf{TE}}/2$$

and

$$\langle \mathbf{q}_i, \mathbf{k}_j \rangle = \eta \langle \mathbf{p}_i^{\mathsf{TE}}, \mathbf{p}_{j+\ell}^{\mathsf{TE}} \rangle \leq \eta d_{\mathsf{TE}}/2 - \eta \varepsilon_T, \ \forall j \neq i - \ell.$$

where $\varepsilon_T > 0$. This implies $\langle \mathbf{q}_i, \mathbf{k}_j \rangle$ is maximized when $j = i - \ell$ with a non-zero gap $\eta \varepsilon_T$, and therefore, we have

$$s_{i,i-\ell} = \frac{\exp(\langle \mathbf{q}_i, \mathbf{k}_{i-\ell} \rangle)}{\sum_{j \in [i]} \exp(\langle \mathbf{q}_i, \mathbf{k}_j \rangle)} \geq \frac{\exp(\eta \varepsilon_T)}{\exp(\eta \varepsilon_T) + (i-1)}. \tag{7}$$

Now we consider the case where $\langle \tilde{\mathbf{h}}_i, \tilde{\mathbf{u}}_{<\mathsf{x}>} \rangle = 0$. Again, by (6) and the orthonormal assumption of token embeddings, we have

$$\langle \tilde{\mathbf{h}}_i, \tilde{\mathbf{u}}_{<\bar{\mathsf{x}}>} \rangle = \sum_{v \in \mathsf{Voc} \backslash \{<\mathsf{x}>\}} \langle \tilde{\mathbf{h}}_i, \tilde{\mathbf{u}}_v \rangle = \sum_{v \in \mathsf{Voc} \backslash \{<\mathsf{x}>\}} \lambda_{i,v} \geq \sum_{v \in \mathsf{Voc} \backslash \{<\mathsf{x}>\}} \lambda_{i,v}^2 = 1.$$

Note that for any $j \in \{2, \ldots, i\}$,

$$\langle \mathbf{p}_{i-j-T}^{\widetilde{\mathsf{TE}}}, \mathbf{p}_0^{\widetilde{\mathsf{TE}}} \rangle = \cos((i-j-T)\omega^{d/2})$$

$$= \cos \left( \frac{T - (i-j)}{M} \right)$$

$$< \cos \left( \frac{T - (i-1)}{M} \right)$$

$$= \langle \mathbf{p}_{i-1-T}^{\widetilde{\mathsf{TE}}}, \mathbf{p}_0^{\widetilde{\mathsf{TE}}} \rangle,$$

where the inequality holds due to $M \geq T$ and the cosine function decreases strictly in $[0, 1]$.

Then, we can define $\xi = \max_{1 \leq j < i \leq T} \frac{2d_{\mathsf{PE}}}{\langle \mathbf{p}_{i-1-T}^{\widetilde{\mathsf{TE}}}, \mathbf{p}_0^{\widetilde{\mathsf{TE}}} \rangle - \langle \mathbf{p}_{i-j-T}^{\widetilde{\mathsf{TE}}}, \mathbf{p}_0^{\widetilde{\mathsf{TE}}} \rangle}$, and thus

$$\xi \left( \langle \mathbf{p}_{i-1-T}^{\widetilde{\mathsf{TE}}}, \mathbf{p}_0^{\widetilde{\mathsf{TE}}} \rangle - \langle \mathbf{p}_{i-j-T}^{\widetilde{\mathsf{TE}}}, \mathbf{p}_0^{\widetilde{\mathsf{TE}}} \rangle \right) \geq 2d_{\mathsf{PE}}, \quad \forall 1 \leq j < i \leq T.$$

Note that for any $j \in \{2, \ldots, T\}$, we have

$$
\begin{aligned}
&\langle \mathbf{q}_i, \mathbf{k}_1 \rangle - \langle \mathbf{q}_i, \mathbf{k}_j \rangle \\
=&\eta \left( \xi \langle \tilde{\mathbf{u}}_{<\bar{\mathbf{x}}>}, \tilde{\mathbf{h}}_i \rangle \left( \langle \mathbf{p}_{i-1-T}^{\widetilde{\mathsf{TE}}}, \mathbf{p}_0^{\widetilde{\mathsf{TE}}} \rangle - \langle \mathbf{p}_{i-j-T}^{\widetilde{\mathsf{TE}}}, \mathbf{p}_0^{\widetilde{\mathsf{TE}}} \rangle \right) - \left( \langle \bar{\mathbf{p}}_i, \bar{\mathbf{p}}_{j+\ell} \rangle - \langle \bar{\mathbf{p}}_i, \bar{\mathbf{p}}_{1+\ell} \rangle \right) \right) \\
\geq& \eta \left( 2d_{\mathsf{PE}} - d_{\mathsf{PE}} \right) \\
=&\eta d_{\mathsf{PE}}.
\end{aligned}
$$

Therefore,

$$s_{i,1} = \frac{\exp(\langle \mathbf{q}_i, \mathbf{k}_1 \rangle)}{\sum_{j \in [i]} \exp(\langle \mathbf{q}_i, \mathbf{k}_j \rangle)} \geq \frac{\exp(\eta d_{\mathsf{PE}})}{\exp(\eta d_{\mathsf{PE}}) + (i - 1)}. \tag{8}$$

By choosing a sufficiently large $\eta$, the lower bound of (7) and (8) will both exceed $1 - \varepsilon$ and thus the proof is complete. $\qquad\square$

## C  Experiment Details

### C.1  Dataset

Table 4: ProsQA statistics. Numbers are averaged over problem instances.

|       | #Problems | $|V|$ | $|E|$ | Sol. Len. |
|-------|-----------|-------|-------|-----------|
| Train | 14785     | 22.8  | 36.5  | 3.5       |
| Val   | 257       | 22.7  | 36.3  | 3.5       |
| Test  | 419       | 22.7  | 36.0  | 3.5       |

The statistics of the ProsQA dataset is shown in Table 4.

### C.2  Stability of the Representation

Table 5: The inner products between $i$-th continuous thoughts and nodes (3 runs with different random seeds).

|               | Step 1             | Step 2            | Step 3            | Step 4            |
|---------------|--------------------|-------------------|-------------------|-------------------|
| Not Reachable | $-0.37, -0.25, -0.33$ | $-0.26, -0.04, -0.14$ | $-0.09, -0.01, 0.02$ | $-0.25, -0.23, -0.27$ |
| Reachable     | $3.59, 3.62, 3.71$ | $1.55, 1.42, 1.37$ | $0.80, 0.77, 0.62$ | $0.61, 0.66, 0.53$ |
| –Frontier     | $5.09, 5.13, 5.38$ | $2.69, 2.45, 2.63$ | $2.11, 1.95, 2.01$ | $2.27, 2.12, 2.29$ |
| –Optimal      | $6.41, 6.52, 6.84$ | $4.78, 4.67, 5.11$ | $6.00, 5.44, 6.43$ | $9.48, 8.98, 9.58$ |

To test whether COCONUT can always learn the desired superpositional search behavior, we conduct multiple experiments with 3 random seeds. We report the mean inner products between continuous thoughts and each node group in Table 5, following the setting in Figure 6. The results are consistent across multiple runs.

### C.3  Computing Resources

Each run of COCONUT takes about 24 hours on two Nvidia A100 80GB GPUs.

