# OpenReview forum: "Reasoning by Superposition: A Theoretical Perspective on Chain of Continuous Thought"
_NeurIPS.cc/2025/Conference — NeurIPS 2025 poster_

### Official Review · Reviewer_5d3n · 2025-06-23

**Clarity:** 4
**Significance:** 3
**Originality:** 4
**Rating:** 4
**Confidence:** 3

**Summary:**

This paper provides a theoretical analysis and experiments on directed graph reachability to explain why Chain of Continuous (Coconut) Thought outperforms Chain of Thought (CoT, the discrete counterpart). The core idea is that CoT is working in a step-by-step manner; the thinking process can only contain one single reasoning path at a time, while Coconut implicitly traverses multiple reasoning paths in parallel within the continuous thought vector. The paper proves that a two-layer transformer with $D$ steps of continuous CoTs can solve the directed graph reachability problem of a graph with diameter $D$. The paper also provides experiments that show that a 2-layer transformer outperforms a 12-layer Coconut transformer with CoT.

**Questions:**

1. As described in the second weakness, can the viewpoint of this paper be generalized to more complex tasks?
2. Do you need to increase the capacity of the continuous thought when the task gets more complex?
3. In the graph reachability problem, can the capacity of the continuous thought reduce the steps needed to get the answer?
4. As described the the third weakness, how would that affect your conclusions?

**Ethical Concerns:**

["NO or VERY MINOR ethics concerns only"]

**Final Justification:**

The authors have addressed some of my concerns, but my main concern remains about the practical value and applicability of the proposed approach for general-purpose reasoning in real-world scenarios. I will keep my score.

**Limitations:**

yes

**Quality:**

4

**Strengths And Weaknesses:**

**Strength**:
1. The paper is well-written and well-motivated -- trying to understand why Coconut outperforms discrete CoT.
2. The paper proposes an interesting perspective: Coconut performs multiple reasoning paths in parallel, and tries to prove it on the graph reachability task.
3. The experimental results, including ablation studies, support the theoretical analysis.

**Weakness**:
1. The paper focuses solely on synthetic graph reachability tasks; it is unclear how well these proofs and findings can be generalized to most other natural language problems that LLMs face. While the paper aims to answer why Coconut outperforms its discrete counterparts in various reasoning tasks, however, this paper solely focuses on the graph problem and how the advantage can be generalized to other tasks are not discussed.
2. The continuous thought vectors may saturate in more complex tasks. The graph reachability task is a relatively easy task where a simple reasoning path can be easily represented, and the inference of the next step is also easy (just traverse). However, when the task gets complicated, for example, the math reasoning problem. The continuous thought vectors may not be able to carry many more reasoning paths than the written thoughts in CoT.
3. The theory part of this paper assumes that the token embeddings are orthonormal; however, this is not true in general LLM's embedding space. [1]

[1] Mikolov, Tomas, et al. "Efficient estimation of word representations in vector space." arXiv preprint arXiv:1301.3781 (2013).

---

> ### Author Rebuttal · Authors · 2025-07-31
>
> We thank the reviewer for the insightful and helpful comments. Below are our responses.
>
> >1. As described in the second weakness, can the viewpoint of this paper be generalized to more complex tasks?
>
> We argue that our results can actually be applied to more general and complex settings:
>
>  (1) **Graph reachability is already general and can be extended to more general problems such as the Turing machine halting problem.** Note that the graph reachability problem can capture general logical relations, and is an abstraction of many real-world problems involving planning, searching, logic deduction, etc. Specifically, it captures one of the most fundamental challenges in reasoning problems - deciding the correct branch among many plausible candidates. Moreover, our theoretical construction for graph reachability can be generalized to more general settings, such as Turing machines, where states of a Turing machine can be modeled as nodes in the graph and transition functions can be modeled as edges. Therefore, determining whether a state is reachable can be converted to determining whether a node can be reached in a directed graph.
>
> (2) Beyond graph reachability (which includes tasks that require planning), **our framework can also be extended to other domains such as arithmetic reasoning.**  For example, a recent theoretical work [6] showed that continuous thought also helps solve the minimum non-negative sum problem by parallel exploration and superposition, where the i-th thought can be viewed as a superposition of all possible (signed) sums generated by the first i numbers.
>
> (3) **Experimental results in real-world benchmarks conducted by [7] also show that continuous thought exhibits superposition** when solving math problems.  For example, when solving a GSM-8k problem, the first continuous thought can be decoded into tokens like “180”, “ 180” (with a space), and “9” where the corresponding reasoning trace could be 3 × 3 × 60 = 9 × 60 = 540, or 3 × 3 × 60 = 3 × 180 = 540. This is exactly a superposition of possible intermediate variables in the calculation, which corresponds to two possible reasoning traces.
>
> (4) Finally, we emphasize that **continuous thought can benefit reasoning in different ways** (e.g., might reduce the number of decoding steps since it does not need to output language tokens that are merely for language coherence), and **superposition is one of the most important mechanisms that we aim to study in this paper.**  Note that superposition is most salient and helpful in problems with large branching factors and requiring extensive search, and the graph reachability task captures this feature very well. We can anticipate that for tasks where the bottleneck goes beyond choosing the correct path among many possible candidates, continuous thoughts might benefit reasoning in other ways (such as reducing the number of decoding steps since it does not need to output language tokens only for coherence as mentioned above), instead of mainly relying on superposition.
>
> > 2. Do you need to increase the capacity of the continuous thought when the task gets more complex?
>
> According to our theory, the only requirement for the capacity of continuous thought is that the embedding dimension should be linear in the number of vertices of the graph, which is a relatively mild requirement and common practice used in many previous theoretical works (e.g., [1-5]).
>
> > 3. In the graph reachability problem, can the capacity of the continuous thought reduce the steps needed to get the answer?
>
> Yes. In principle, if we allow the embedding dimension to be extremely large (e.g., exponential in graph size), it might even be possible to use only a constant number of steps to get the answer. In our paper, we use the common assumption that the embedding dimension is linear in the number of vertices of the graph (e.g., [1-5]), which is more practical.
>
> > 4. As described the the third weakness, how would that affect your conclusions?
>
> Our theory actually does not require that token embeddings are exact orthonormal, but only need to be approximately orthogonal as in practical scenarios. For example, If we assume that for two different nodes, the inner product of their embeddings are $\varepsilon$ (e.g., $\varepsilon = 0.1$) instead of 0 which is the ideal exact orthonormal setting, our theorem still holds since the attention mechanism can downweight the noise through softmax operation, and MLP will filter out the noise and keep only the useful signals. Our theoretical construction also works for the case where the inner product of different token embeddings is less than or equal to $\varepsilon$ with a smaller $\varepsilon$ and more fine-grained analysis. We adopted the orthonormal assumption to make the analysis and main message cleaner, and we will add the discussion and analysis of the more practical approximate orthogonal setting in the revision. We thank the reviewer for the suggestion.
>
> **References:**
>
> [1] Tian Y, Wang Y, Chen B, Du SS. Scan and snap: Understanding training dynamics and token composition in 1-layer transformer. Advances in neural information processing systems. 2023 Dec 15;36:71911-47.
>
> [2] Tian Y, Wang Y, Zhang Z, Chen B, Du S. Joma: Demystifying multilayer transformers via joint dynamics of mlp and attention. arXiv preprint arXiv:2310.00535. 2023 Oct 1.
>
> [3] Chen L, Bruna J, Bietti A. Distributional associations vs in-context reasoning: A study of feed-forward and attention layers. arXiv preprint arXiv:2406.03068. 2024 Jun 5.
>
> [4] Nichani E, Damian A, Lee JD. How transformers learn causal structure with gradient descent. arXiv preprint arXiv:2402.14735. 2024 Feb 22.
>
> [5] Nguyen Q, Nguyen-Tang T. One-Layer Transformers are Provably Optimal for In-context Reasoning and Distributional Association Learning in Next-Token Prediction Tasks. arXiv preprint arXiv:2505.15009. 2025
> May 21.
>
> [6] Gozeten HA, Ildiz ME, Zhang X, Harutyunyan H, Rawat AS, Oymak S. Continuous Chain of Thought Enables Parallel Exploration and Reasoning. arXiv preprint arXiv:2505.23648. 2025 May 29.
>
> [7] Hao S, Sukhbaatar S, Su D, Li X, Hu Z, Weston J, Tian Y. Training large language models to reason in a continuous latent space. arXiv preprint arXiv:2412.06769. 2024 Dec 9.

---

### Official Review · Reviewer_BPy1 · 2025-07-01

**Clarity:** 4
**Significance:** 3
**Originality:** 4
**Rating:** 5
**Confidence:** 3

**Summary:**

This paper provides theoretical guarantees for this paper: *Training Large Language Models to Reason in a Continuous Latent Space.*

**Questions:**

According to Lemma 2, it seems the model must extend its reasoning one more “hop” in the graph at each step. To keep training stable, if we move a portion of the language-based reasoning into the latent space at every stage, doesn’t that mean that what used to be a single fine-tuning pass now requires many rounds? If I’ve understood correctly, wouldn’t this greatly increase the overall training cost?

**Ethical Concerns:**

["NO or VERY MINOR ethics concerns only"]

**Final Justification:**

I am keeping my score unchanged.

**Limitations:**

yes

**Quality:**

4

**Strengths And Weaknesses:**

Strengths: This paper offers rigorous theoretical proofs and simulation experiments that demonstrate its effectiveness on reasoning tasks such as graph reachability. Although the article focuses on graph reachability as a case study, its theoretical analysis and constructions—enabling “relative‐offset” access—apply to any task, with the potential to extend to path search, planning, theorem proving, and other complex reasoning scenarios.

I didn't find any obvious shortcomings in this paper.

---

> ### Author Rebuttal · Authors · 2025-07-31
>
> We thank the reviewer for the insightful and helpful comments. Below are our responses.
>
> >According to Lemma 2, it seems the model must extend its reasoning one more “hop” in the graph at each step. To keep training stable, if we move a portion of the language-based reasoning into the latent space at every stage, doesn’t that mean that what used to be a single fine-tuning pass now requires many rounds? If I’ve understood correctly, wouldn’t this greatly increase the overall training cost?
>
>
> Yes, the training cost is indeed higher compared to the cost of discrete thought since it involves multiple training stages. One possible method to reduce the training cost is to convert multiple steps (or tokens) from the language space to the latent continuous space at each stage, as in the original coconut paper [1]. However, there might be a tradeoff between training efficiency and stability, and we need to choose the number of steps/tokens moving from the language space to the latent space at each stage appropriately.  Although it is beyond the scope of this paper to design more efficient training methods in continuous space, we agree with the reviewer that it is a very important future direction.
>
>
> **References:**
>
> [1] Hao S, Sukhbaatar S, Su D, Li X, Hu Z, Weston J, Tian Y. Training large language models to reason in a continuous latent space. arXiv preprint arXiv:2412.06769. 2024 Dec 9.

---

> > ### Comment · Reviewer_BPy1 · 2025-08-01
> >
> > Thank you for the clarification; I am keeping my score unchanged.

---

### Official Review · Reviewer_wcdH · 2025-07-03

**Clarity:** 3
**Significance:** 2
**Originality:** 3
**Rating:** 4
**Confidence:** 1

**Summary:**

I am not very familiar with the underlying theoretical framework used in this paper. In my view, this paper develops a theoretical construction showing that two-layer transformers with continuous latent chains can solve directed graph reachability in a number of steps equal to the graph’s diameter, while discrete chains require quadratically more steps. The proofs use practical positional encodings, and experiments on synthetic and math tasks dshow similar behaviors emerge during training.

**Questions:**

- How practically useful is this theory for large-scale LLMs?
- Can the authors discuss in more details on how the proposed construction transfer to other domains beyond graph reachability?

**Ethical Concerns:**

["NO or VERY MINOR ethics concerns only"]

**Final Justification:**

The authors have addressed the questions, and I would maintain my score.

**Limitations:**

Yes

**Quality:**

3

**Strengths And Weaknesses:**

---
**Quality (3)**
- Strengths:
    - Clear separation result between continuous and discrete chain-of-thought expressivity.
    - Proofs account for realistic positional encodings (sinusoidal, RoPE).
    - Empirical results align tightly with theory, with useful visualizations of latent states.
- Weaknesses:
    - Relies on idealized assumptions (exact orthonormal embeddings, large embedding dimension).
    - Requires embedding size proportional to vocabulary, which may be impractical in real LLMs.

---

**Clarity (3)**
- Strengths: Overall, the paper is well formatted and has a clear introduction and literature review.
- Weaknesses:
    - Heavy notation and many indices make the paper hard to parse for someone new to transformer expressivity.
    - Key mechanisms (attention chooser, buffer/MLP filter) could be summarized more succinctly, with full details in the appendix.

---

**Significance (2)**
- Strengths: in general, understanding why (continuous) latent reasoning helps with theoretical insights is an important topic and can inspire many future works.
- Weaknesses:
    - The paper focused on a toy reachability problem. It is slightly unclear on how the proposed analysis can transfer to real-world reasoning tasks or larger LLMs.
    - Practical impact on real world benchmarks remains to be validated.

---

**Originality (3)**
- Strengths: I believe the  theoretical link between continuous superposition and expressivity speedup is fresh.

---

---

> ### Author Rebuttal · Authors · 2025-07-31
>
> We thank the reviewer for the insightful and helpful comments. Below are our responses.
>
> > Weaknesses (Quality): Relies on idealized assumptions (exact orthonormal embeddings, large embedding dimension).
>     Requires embedding size proportional to vocabulary, which may be impractical in real LLMs.
>
> It is a very common practice that embedding size is proportional to vocabulary in theoretical analysis such as previous works [1-5]. Note that this is a natural consequence of the assumption of orthonormal embeddings. While all previous work [1-5] assumed orthonormal embeddings (or even one-hot embeddings), our theory actually does not require that token embeddings are exact orthonormal, but only need to be approximately orthogonal as in practical scenarios. For example, If we assume that for two different nodes, the inner product of their embeddings are $\varepsilon$ (e.g., $\varepsilon = 0.1$) instead of 0 which is the ideal exact orthonormal setting, our theorem still holds since the attention mechanism can downweight the noise through softmax operation, and MLP will filter out the noise and keep only the useful signals. Our theoretical construction also works for the case where the inner product of different token embeddings is less than or equal to $\varepsilon$ with a smaller $\varepsilon$ and more fine-grained analysis. We adopted the orthonormal assumption to make the analysis and main message cleaner, and we will add the discussion and analysis of the more practical approximate orthogonal setting in the revision. We thank the reviewer for the suggestion.
>
> > Weaknesses (Clarity):
> Heavy notation and many indices make the paper hard to parse for someone new to transformer expressivity.
> Key mechanisms (attention chooser, buffer/MLP filter) could be summarized more succinctly, with full details in the appendix.
>
> We thank the reviewer for the suggestions in writing. We will utilize the additional content page in the camera-ready version to make notations easier to read and key mechanisms more succinct, and add more details in the appendix.
>
> > Weaknesses (Significance):
> The paper focused on a toy reachability problem. It is slightly unclear on how the proposed analysis can transfer to real-world reasoning tasks or larger LLMs.
> Practical impact on real world benchmarks remains to be validated.
>
> See responses to the second question.
>
> > Question 1: How practically useful is this theory for large-scale LLMs?
>
> Previous work shows experimental results that continuous thought outperforms discrete thought in certain types of reasoning tasks, but did not provide a fine-grained analysis of why continuous thoughts can reason more efficiently and what types of tasks benefit more from continuous thought. Our theory identifies and provides a rigorous analysis of one of the most important mechanisms, i.e., superpositions that enable parallel search, which makes continuous thoughts powerful. According to our analysis, superposition is more helpful for problems with large branching factors and requiring extensive search. Therefore, our theory can guide large-scale LLMs on whether to choose continuous thoughts to solve a given task.
>
> > Question 2: Can the authors discuss in more details on how the proposed construction transfer to other domains beyond graph reachability?
>
> Below, we discuss how our results can be applied to more general and complex settings:
>
> (1) **Graph reachability is already general and can be extended to more general problems such as the Turing machine halting problem.** Note that the graph reachability problem can capture general logical relations, and is an abstraction of many real-world problems involving planning, searching, logic deduction, etc. Specifically, it captures one of the most fundamental challenges in reasoning problems - deciding the correct branch among many plausible candidates. Moreover, our theoretical construction for graph reachability can be generalized to more general settings, such as Turing machines, where states of a Turing machine can be modeled as nodes in the graph and transition functions can be modeled as edges. Therefore, determining whether a state is reachable can be converted to determining whether a node can be reached in a directed graph.
>
> (2) Beyond graph reachability (which includes tasks that require planning), **our framework can also be extended to other domains such as arithmetic reasoning.**  For example, a recent theoretical work [6] showed that continuous thought also helps solve the minimum non-negative sum problem by parallel exploration and superposition, where the i-th thought can be viewed as a superposition of all possible (signed) sums generated by the first i numbers.
>
> (3) **Experimental results in real-world benchmarks conducted by [7] also show that continuous thought exhibits superposition** when solving math problems.  For example, when solving a GSM-8k problem, the first continuous thought can be decoded into tokens like “180”, “ 180” (with a space), and “9” where the corresponding reasoning trace could be 3 × 3 × 60 = 9 × 60 = 540, or 3 × 3 × 60 = 3 × 180 = 540. This is exactly a superposition of possible intermediate variables in the calculation, which corresponds to two possible reasoning traces.
>
> (4) Finally, we emphasize that **continuous thought can benefit reasoning in different ways** (e.g., might reduce the number of decoding steps since it does not need to output language tokens that are merely for language coherence), and **superposition is one of the most important mechanisms that we aim to study in this paper.**  Note that superposition is most salient and helpful in problems with large branching factors and requiring extensive search, and the graph reachability task captures this feature very well. We can anticipate that for tasks where the bottleneck goes beyond choosing the correct path among many possible candidates, continuous thoughts might benefit reasoning in other ways (such as reducing the number of decoding steps since it does not need to output language tokens only for coherence as mentioned above), instead of mainly relying on superposition.
>
> **References:**
>
> [1] Tian Y, Wang Y, Chen B, Du SS. Scan and snap: Understanding training dynamics and token composition in 1-layer transformer. Advances in neural information processing systems. 2023 Dec 15;36:71911-47.
>
> [2] Tian Y, Wang Y, Zhang Z, Chen B, Du S. Joma: Demystifying multilayer transformers via joint dynamics of mlp and attention. arXiv preprint arXiv:2310.00535. 2023 Oct 1.
>
> [3] Chen L, Bruna J, Bietti A. Distributional associations vs in-context reasoning: A study of feed-forward and attention layers. arXiv preprint arXiv:2406.03068. 2024 Jun 5.
>
> [4] Nichani E, Damian A, Lee JD. How transformers learn causal structure with gradient descent. arXiv preprint arXiv:2402.14735. 2024 Feb 22.
>
> [5] Nguyen Q, Nguyen-Tang T. One-Layer Transformers are Provably Optimal for In-context Reasoning and Distributional Association Learning in Next-Token Prediction Tasks. arXiv preprint arXiv:2505.15009. 2025 May 21.
>
> [6] Gozeten HA, Ildiz ME, Zhang X, Harutyunyan H, Rawat AS, Oymak S. Continuous Chain of Thought Enables Parallel Exploration and Reasoning. arXiv preprint arXiv:2505.23648. 2025 May 29.
>
> [7] Hao S, Sukhbaatar S, Su D, Li X, Hu Z, Weston J, Tian Y. Training large language models to reason in a continuous latent space. arXiv preprint arXiv:2412.06769. 2024 Dec 9.

---

### Official Review · Reviewer_X4Dq · 2025-07-03

**Clarity:** 3
**Significance:** 4
**Originality:** 4
**Rating:** 5
**Confidence:** 4

**Summary:**

This paper gives theoretical investigation of the expressivity and reasoning capability of LLMs when using continuous chains of thought (CoT), specifically, the COCONUT framework. The authors focus on the graph reachability problem and provide a theoretical construction showing that a two-layer transformer with D continuous latent steps (where D is the diameter of the graph) can solve the problem efficiently. This contrasts with discrete CoT methods, which require O(n^2) steps for graphs with n nodes. The key finding is that continuous latent thoughts act as "superposition states", representing multiple search frontiers simultaneously—akin to a parallel BFS—while discrete CoTs sequentially explore paths, leading to inefficiency and possible local minima.

**Questions:**

1. Generalization beyond graph reachability
While the theoretical framework is very interesting, it focuses on the setup of graph reachability. Do you anticipate similar superposition-based representations to emerge for reasoning tasks that are not inherently graph-structured (e.g., algebraic reasoning, program synthesis)? Can your theoretical framework be extended or adapted to such domains?

2. Superposition representation robustness
How robust are the emergent superposition representations to noise, adversarial graph perturbations, or imperfect supervision? Have you evaluated how the representations evolve when the graph has spurious or misleading edges?

3. Interpretability of the continuous reasoning path
While the superposition representation appears to be powerful, how interpretable is it in practice? Can individual paths or reasoning traces be extracted or disentangled from the latent vectors?

**Ethical Concerns:**

["NO or VERY MINOR ethics concerns only"]

**Final Justification:**

Overall a solid paper that definitely deserves more visibility and broader discussion.

**Limitations:**

Yes

**Quality:**

4

**Strengths And Weaknesses:**

Strengths
Novel theoretical contribution: The work formalizes the intuition behind continuous CoT's advantage via a compelling superposition-based analogy and mathematical proof, which is a very innovative way of analyzing reasoning behaviors and has deep implications.

Weaknesses
Limited scope: The paper focuses specifically on the graph reachability problem, and While the problem is a representative benchmark, the theoretical findings are not yet generalized to broader reasoning tasks (e.g. arithmetic, planning, theorem proving).

---

> ### Author Rebuttal · Authors · 2025-07-31
>
> We thank the reviewer for the insightful and helpful comments. Below are our responses.
>
> > 1. Generalization beyond graph reachability While the theoretical framework is very interesting, it focuses on the setup of graph reachability. Do you anticipate similar superposition-based representations to emerge for reasoning tasks that are not inherently graph-structured (e.g., algebraic reasoning, program synthesis)? Can your theoretical framework be extended or adapted to such domains?
>
> We argue that our results can actually be applied to more general and complex settings:
>
>  (1) **Graph reachability is already general and can be extended to more general problems such as the Turing machine halting problem.** Note that the graph reachability problem can capture general logical relations, and is an abstraction of many real-world problems involving planning, searching, logic deduction, etc. Specifically, it captures one of the most fundamental challenges in reasoning problems - deciding the correct branch among many plausible candidates. Moreover, our theoretical construction for graph reachability can be generalized to more general settings, such as Turing machines, where states of a Turing machine can be modeled as nodes in the graph and transition functions can be modeled as edges. Therefore, determining whether a state is reachable can be converted to determining whether a node can be reached in a directed graph.
>
> (2) Beyond graph reachability (which includes tasks that require planning), **our framework can also be extended to other domains such as arithmetic reasoning.**  For example, a recent theoretical work [1] showed that continuous thought also helps solve the minimum non-negative sum problem by parallel exploration and superposition, where the i-th thought can be viewed as a superposition of all possible (signed) sums generated by the first i numbers.
>
> (3) **Experimental results in real-world benchmarks conducted by [2] also show that continuous thought exhibits superposition** when solving math problems.  For example, when solving a GSM-8k problem, the first continuous thought can be decoded into tokens like “180”, “ 180” (with a space), and “9” where the corresponding reasoning trace could be 3 × 3 × 60 = 9 × 60 = 540, or 3 × 3 × 60 = 3 × 180 = 540. This is exactly a superposition of possible intermediate variables in the calculation, which corresponds to two possible reasoning traces.
>
> (4) Finally, we emphasize that **continuous thought can benefit reasoning in different ways** (e.g., might reduce the number of decoding steps since it does not need to output language tokens that are merely for language coherence), and **superposition is one of the most important mechanisms that we aim to study in this paper.**  Note that superposition is most salient and helpful in problems with large branching factors and requiring extensive search, and the graph reachability task captures this feature very well. We can anticipate that for tasks where the bottleneck goes beyond choosing the correct path among many possible candidates, continuous thoughts might benefit reasoning in other ways (such as reducing the number of decoding steps since it does not need to output language tokens only for coherence as mentioned above), instead of mainly relying on superposition.
>
> > 2. Superposition representation robustness How robust are the emergent superposition representations to noise, adversarial graph perturbations, or imperfect supervision? Have you evaluated how the representations evolve when the graph has spurious or misleading edges?
>
> Note that our theoretical constructions work for any directed graph (even a graph containing cycles). This means our main theorem holds even for the worst case, and thus our construction is robust to spurious or misleading edges.
>
> > 3. Interpretability of the continuous reasoning path While the superposition representation appears to be powerful, how interpretable is it in practice? Can individual paths or reasoning traces be extracted or disentangled from the latent vectors?
>
> **Interpretability of the continuous thoughts:** Note that the continuous thought is a superposition of reachable nodes, so it can be interpreted as a probability distribution over different tokens. As in our experiments, one can calculate the inner product of the current thought and different token embeddings (and possibly apply a softmax operation) to obtain a distribution over different nodes. Beyond the graph reachability problem we focused on in our paper, the superposition representation is also interpretable in real-world datasets. For example, when solving a GSM-8k problem, the experiments in [2] shows that the first continuous thought can be decoded into tokens like “180”, “ 180” (with a space), and “9” where the corresponding reasoning trace could be 3 × 3 × 60 = 9 × 60 = 540, or 3 × 3 × 60 = 3 × 180 = 540. The interpretations of the first thought happen to be the first intermediate variables in the calculation.
>
> **Decoding individual reasoning traces:** First, we want to emphasize that the continuous thought stores the reasoning paths very efficiently. According to our construction, the continuous thoughts only store all reachable nodes (and thus all frontier nodes), but not all trajectories. Note that there may exist an exponential number of trajectories for a given graph and a start-destination node pair (e.g., a 2-dimensional grid graph with k*k nodes. Then, for the start node (1,1) and the destination node (k,k), there can be C(2k-2,k-1) paths which is exponential in k). By only storing all frontier nodes, continuous thoughts **implicitly** store an exponential number of reasoning traces. This is in stark contrast to the discrete CoT, which can only **explicitly** store one reasoning trace and is much less efficient.  Since continuous thoughts maintain all possible reasoning traces, one should not anticipate extracting an individual path in the middle of the reasoning process. However, after the model obtains the final answer, one can use it to collapse the superposition of the continuous thought and extract the desired individual reasoning trace. For example, in our graph reachability task, given the final answer (the reachable destination node), one can decode the path using stored frontier nodes step-by-step “from right to left”.
>
> To decode the reasoning trace for more general tasks, one possible method is to train a VAE (variational auto-encoder) to decode the continuous thoughts to obtain the discrete thoughts. As discussed above, the VAE should also receive the final answer as the input to extract a ``collapsed’’ reasoning trace conditioned on the final answer. Although it is beyond the scope of this paper to design algorithms to extract the reasoning traces for a given task, it is a very interesting and important future direction.
>
> **Reference:**
>
> [1] Gozeten HA, Ildiz ME, Zhang X, Harutyunyan H, Rawat AS, Oymak S. Continuous Chain of Thought Enables Parallel Exploration and Reasoning. arXiv preprint arXiv:2505.23648. 2025 May 29.
>
> [2] Hao S, Sukhbaatar S, Su D, Li X, Hu Z, Weston J, Tian Y. Training large language models to reason in a continuous latent space. arXiv preprint arXiv:2412.06769. 2024 Dec 9.

---

> > ### Comment · Reviewer_X4Dq · 2025-08-08
> >
> > Thank you authors for your very detailed and thoughtful answers, they are very well-written. I like this paper, we need more of such papers that go beyond simple engineering tricks and provides deeper theoretical insights into the inner working of LLMs. I also believe that verbal COT will eventually be replaced by continuous versions due to tighter information compression ratio. This paper is also potentially linked to the argument that consciousness is a superposition of quantum states within brain.

---

### Note · Authors · 2025-08-11

We thank the reviewers for all the efforts in reviewing our paper and providing helpful and insightful feedback! Below we summarize the rebuttal content to faciliate AC's final decision.

1. We are happy to see that all reviewers are very positive about our work in the review, especially the following points:

- Our paper is well-written and well-motivated, formalizing the intuition behind continuous CoT's advantage via a compelling superposition-based analogy and rigorous mathematical proof, which is a very innovative way of analyzing reasoning behaviors and has deep implications.
- Our proofs account for realistic positional encodings (sinusoidal, RoPE).
- Our experimental results align tightly with our theory, with useful visualizations of latent states.



2. In the response, we have clarified all reviewer concerns. Below we summarize the main points:

 -  Generalization beyond graph reachability: We provided a very detailed response and summarized our main points as follows. (1) Graph reachability is already general and can be extended to more general problems such as the Turing machine halting problem. (2) Our framework can also be extended to other domains such as arithmetic reasoning. (3) Experimental results in real-world benchmarks in previous work also show that continuous thought exhibits superposition when solving math problems. (4) Continuous thought can benefit reasoning in different ways, and superposition is one of the most important mechanisms that we aim to study in this paper.
 - Interpretability and decoding of the reasoning path: The continuous thought can be interpreted as a probability distribution over different tokens in both synthetic and real-world tasks. Also, by only storing all frontier nodes, continuous thoughts implicitly store an exponential number of reasoning traces, and one can extract the desired individual reasoning trace using the final answer.
 - Orthonormal assumption of embeddings: The orthonormal assumption can be relaxed to approximate orthonormal as in the practical scenario and we provided details in the rebuttal.


3. Reviewers posted very positive comments after our response and did not have any unaddressed concerns.

- Both reviewers X4Dq and BPy1 responded with a high score of 5.
- Both reviewers 5d3n and wcdH entered the final justification and did not mention any unaddressed concerns, so we believe we have addressed all their concerns and will be grateful if they will raise their score accordingly.

---

### Decision · Program_Chairs · 2025-09-17

**Decision:**

Accept (poster)

**Comment:**

The paper theoretical studies the power of continuous chain of thoughts for graph reachability problem. It provides a theoretical construction of a 2-layer Transformer that can solve the problem in $d$ steps where $d$ is the graph diameter (CoT would require $n^2$ steps where $n$ is the number of vertices.) It does so by encoding multiple search paths as a superposition. This shows the benefit of latent thoughts over discrete CoT based reasoning. Experiments are conducted to validate various aspects of the theory.

Reviewers were quite positive about the paper, highlighting the novel theoretical contribution, the superposition based analogy and the empirical validation. Some concerns were raised regarding the limited scope of the analysis (just the graph reachability problem) to which the authors responded that the problem is general enough to capture many interesting settings. Overall this paper makes a solid contribution towards understanding the strengths and limitations of continuous vs discrete CoT in a reasonably general problem of graph reachability.